# Programmable generation of counterrotating bicircular light pulses in the multi-terahertz frequency range

Kotaro Ogawa [1], Natsuki Kanda [1,2] ✉, Yuta Murotani [1] & Ryusuke Matsunaga [1] ✉

The manipulation of solid states using intense infrared or terahertz light fields is a pivotal area in contemporary ultrafast photonics research. While conventional circular polarization has been well explored, the potential of counterrotating bicircular light remains widely underexplored, despite growing interest in theory. In the mid-infrared or multi-terahertz region, experimental challenges lie in difficulties in stabilizing the relative phase between two-color lights and the lack of available polarization elements. Here, we successfully generated phase-stable counterrotating bicircular light pulses in the 14–39 THz frequency range circumventing the above problems. Employing spectral broadening, polarization pulse shaping with a spatial light modulator, and intra-pulse difference frequency generation leveraging a distinctive angular-momentum selection rule within the nonlinear crystal, we achieved direct conversion from near-infrared pulses into the designed counterrotating bicircular multi-terahertz pulses. Use of the spatial light modulator enables programmable control over the shape, orientation, rotational symmetry, and helicity of the bicircular light field trajectory. This advancement provides a novel pathway for the programmable manipulation of light fields, and marks a significant step toward understanding and harnessing the impact of tailored light fields on matter, particularly in the context of topological semimetals.

Recent advancements in phase-stable light fields and ultrafast spectroscopy with sub-cycle time resolution have ushered in a new era of controlling matter through terahertz ($10^{12}$ s$^{-1}$) or petahertz ($10^{15}$ s$^{-1}$) electric fields. In contrast to the direct-current limit, these high-frequency light fields exhibit the capability to drive electrons within a coherent interaction regime, presenting fertile ground for non-perturbative nonlinear interactions and the artificial design of functionalities in solids[1–6]. The chiral field trajectory induced by circularly polarized light is particularly interesting as it breaks the time-reversal symmetry. The resulting time-periodic rotational motion of electrons can be projected onto an effective band structure in the Floquet picture, leading to a topologically nontrivial phase in solids[7,8]. Additionally, intense circularly polarized light can drive the lattice to induce a magnetic moment in nonmagnetic materials[9,10] and even to achieve magnetization switching[11]. Nonlinear current induced by circularly polarized light has also attracted considerable attention for the topological or geometrical aspect of Bloch electrons[12–15] as well as for its close relation to spintronics[16–19].

Moving beyond simple circularly polarized light, more intricate trajectories of the light electric field vector can be realised by superposing two light fields with different frequencies and opposite helicity. For example, superposition of counterrotating light fields with

[1]The Institute for Solid State Physics, The University of Tokyo, 5-1-5 Kashiwanoha, Kashiwa, Chiba 277-8581, Japan. [2]Ultrafast Coherent Soft X-ray Photonics Research Team, RIKEN Center for Advanced Photonics, RIKEN, 2-1 Hirosawa, Wako, Saitama 351-0198, Japan. ✉e-mail: n-kanda@riken.jp; matsunaga@issp.u-tokyo.ac.jp

frequencies $\omega$ (fundamental) and $2\omega$ (second harmonic) delineate a trefoil- or triangle-like shape of field trajectory (illustrated in Fig. 1a.1 and 1a.2) depending on the ratio of field amplitudes. Such counterrotating bicircular light (BCL) pulses have garnered attention in applications such as high harmonic generation (HHG) in gaseous molecules[20,21], coherent control of ionization[22] to generate circularly polarized light in the extreme ultraviolet or soft-X-ray region[23], and generation of attosecond magnetic field[24]. Three-fold rotational symmetry of a BCL holds potential for molecule orientation control[25] and valley manipulation in honeycomb lattices[26–29]. It is also highly intriguing to apply BCL in topological semimetals that host relativistic fermions with a crossing band structure preserved by crystal symmetry or topology; their nontrivial phases may be further manipulated by intense BCL field[30,31]. Moreover, by adjusting the ratio of two-color frequencies $n_1\omega$ and $n_2\omega$, the BCL field trajectory can be tailored to an arbitrary $C_n$ symmetry, where $n = (n_1 + n_2)/\gcd(n_1, n_2) \geq 3$; $\gcd(n_1, n_2)$ represents the greatest common divisor of the natural numbers $n_1$ and $n_2$. As an example, Fig. 1a.3 illustrates a pentagram-like shape for the case of $2\omega$ and $3\omega$. BCL pulses with a large value of $n$ offer an opportunity for an in-depth understanding of light–matter interactions, elucidating the role of spins in HHG[32] and multipath quantum interference in field ionization[33].

To apply BCL to solid-state physics, a phase-stable intense light source in the multi-terahertz regime (10–70 THz in frequency, 4–30 μm in wavelength, 40–300 meV in photon energy) is crucial[34]. The multi-terahertz light field offers several advantages: (i) it oscillates faster than typical electron-phonon scattering times, facilitating coherent driving by suppressing free-carrier absorption; (ii) it resonates with phonon absorption, enabling nonlinear deformation of the crystal structure; (iii) compared with visible or near-infrared (NIR) pulse, the lower frequency in multi-terahertz pulses helps to obtain a large amplitude of the vector potential $|\mathbf{A}| = |\mathbf{E}|/\omega$ with suppressing unwanted inter-band excitation of electrons, allowing nondestructive interaction using a strong light pulse. However, controllability of polarization is relatively poor in the multi-terahertz region owing to lack of dispersion-free optical elements[35]. Moreover, implementing BCL pulses necessitates a fixed carrier envelope phase (CEP) for each color and a stable relative phase between the two-color light fields[36]; otherwise the field trajectory would rotate during measurement due to jitter. Generating BCL pulses by simply combining two beams in the Mach-Zehnder-like geometry requires intricate feedback controls of the CEP and relative phase. If the fundamental beam is strong enough to generate second harmonic, the phase-stable BCL can be generated within a compact in-line geometry as demonstrated in the visible region[37]. However, this method is limited to the $C_3$-symmetric field trajectory, and would not be applied in the multi-terahertz region where it would be difficult to efficiently generate the second harmonic, sharply filter higher-order harmonics, and prepare an achromatic quarter waveplate.

In this paper, we report the generation of phase-stable BCL pulses in the multi-terahertz regime. A broadband NIR pulse, tailored by a computer-controlled spatial light modulator (SLM) in the $4f$ geometry, is directly converted to the desired multi-terahertz BCL pulse through intra-pulse difference frequency generation (DFG) using a unique polarization selection rule in the nonlinear crystal. We demonstrate that the shape, orientation, rotational symmetry, and helicity of the light field trajectory can be arbitrarily controlled across a broad spectral range (14–39 THz) while maintaining phase stability. Fluctuation in the azimuthal angle of the trajectory is kept as small as 15 mrad. This result opens a new pathway for novel light–matter interactions in condensed matter.

## Results

### Circular polarization selection rule for DFG process

Our approach for BCL pulse generation relies on the polarization selection rule in the wavelength conversion from an NIR pulse to the multi-terahertz pulse. Figure 1b.1 illustrates the energy diagram of DFG for the generation of a left-circularly polarized (LCP) pulse with a $2\omega$

frequency in the nonlinear crystal GaSe with a $C_3$ lattice symmetry. A circularly polarized photon has a (spin) angular momentum $\pm\hbar$, which can be transferred to the electron system through light-matter interaction. A right-circularly polarized (RCP) photon with an $\Omega + 2\omega$ frequency in the NIR pulse virtually excites an electron into a state with an angular momentum of $-\hbar$. If the NIR pulse includes an LCP photon with a $\Omega$ frequency simultaneously, this electron is transferred to the second virtual state with the loss of an angular momentum $+\hbar$. Consequently, the final transition to the ground state has to release an energy of $2\omega(= (\Omega + 2\omega) - \Omega)$ and an angular momentum of $-2\hbar(= -\hbar - (+\hbar))$, which is usually forbidden since a photon alone cannot carry this angular momentum. However, in the case of a $C_3$-symmetric crystal, an excess angular momentum of $\pm 3\hbar$ can be transferred to the lattice[38,39]. By releasing an angular momentum of $-3\hbar$ to the lattice, the second virtual state can emit an LCP photon with an angular momentum $-2\hbar + 3\hbar = +\hbar$ and a frequency $2\omega$. In contrast, if the two NIR photons have the same helicity, the DFG process is forbidden because the second virtual state in this case has exactly the same angular momentum as the ground state. Figure 1b.2 depicts the counterpart process, where the excitation with an LCP photon ($\Omega$) and emission of an RCP photon ($\Omega - \omega$) result in the emission of an RCP photon ($\omega$). We employed the two DFG processes simultaneously in a single NIR pulse by using a common LCP frequency component $\Omega$ in both Fig. 1b.1 and 1b.2. Thus, a multi-terahertz BCL pulse can be directly generated without any waveplate or polarizer in the multi-terahertz region. The three required frequency components in the NIR pulse include $\Omega$ (LCP), $\Omega - \omega_R$ (RCP), and $\Omega + \omega_L$ (RCP), termed $C$-band (central), $L$-band (lower), and $U$-band (upper), respectively.

### Near infrared pulse shaping

The polarization-tailored NIR pulse is crafted through the following procedure: The output from a Yb-based laser with a pulse width of 160 fs undergoes spectral broadening using the multi-plate broadening method[40,41] to encompass the three required frequency bands. The broadened pulse is compressed to ~12 fs through dispersion compensation. Subsequently, this pulse is directed into a polarization pulse shaper composed of an SLM with 640 pixels positioned on the Fourier plane of a $4f$ setup, as depicted in Fig. 1c. In this configuration, we utilize a dual-mask SLM to independently manipulate the spectral phases of components polarized along the azimuthal angles of $\pm 45°$. Let $\phi_A(\Omega)$ and $\phi_B(\Omega)$ denote the spectral phases for frequency $\Omega$ given by the two masks. When horizontally polarized light is incident upon the pulse shaper, the resulting output electric field can be expressed using the Jones vector as follows:

$$E_{\text{out}}(\Omega) = \begin{pmatrix} \cos\eta(\Omega) \\ -i\sin\eta(\Omega) \end{pmatrix} e^{i\phi(\Omega)} E_{\text{in}}(\Omega), \tag{1}$$

where $\eta(\Omega) = (\phi_A(\Omega) - \phi_B(\Omega))/2$ and $\phi(\Omega) = (\phi_A(\Omega) + \phi_B(\Omega))/2$. Changing $\eta(\Omega)$ enables continuous tuning of the ellipticity, $\tan\eta(\Omega)$, from $+1$ (LCP) to $-1$ (RCP). This flexibility is used to control the rotating direction of circular polarization in the three frequency bands. Additionally, the average spectral phase $\phi(\Omega)$ includes the offset phase, group delay, and group delay dispersion, all of which can be set on demand.

In the schematic spectrogram presented in Fig. 1d, the central frequency of the $C$-band (green) is chosen to be $\Omega_0/2\pi = 286$ THz (wavelength of 1050 nm), positioned around the peak of the NIR spectrum. The bandwidth is set to 5 THz to generate an appropriate number of cycles in the multi-terahertz BCL pulse. The $L$-band (red) and $U$-band (blue) are also selected to produce two multi-terahertz pulses with $\omega_L = 2\omega$ and $\omega_R = \omega$, where $\omega/2\pi = 17$ THz. To prevent DFG of undesired frequencies, other NIR frequency components are transformed into RCP and temporally retarded by a 400-fs group delay (Supplementary Information Note 1). The $L$- and $U$-bands possess the same helicity, prohibiting DFG between them (resulting in a frequency

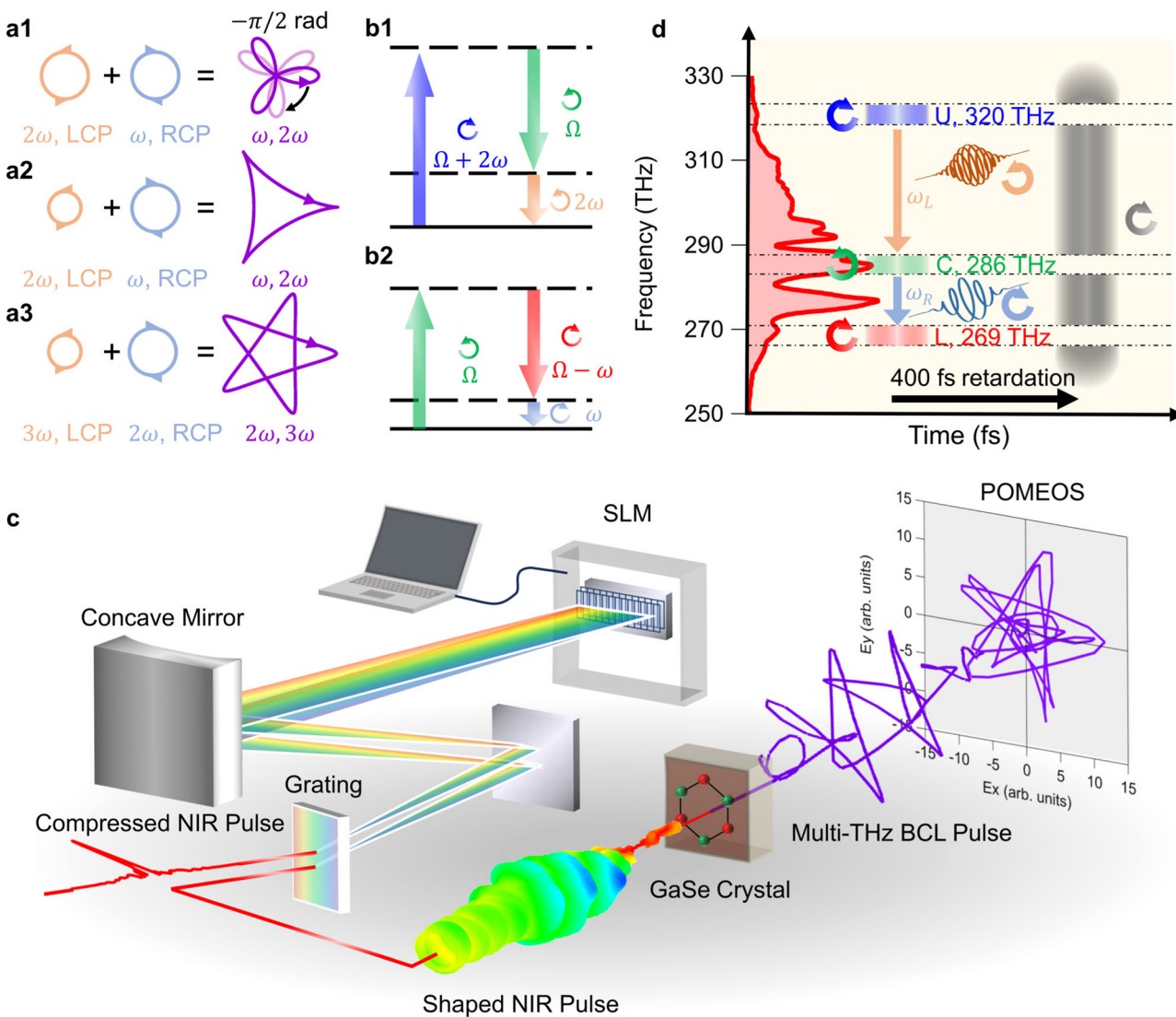

**Fig. 1 | Generation of phase-stable multi-terahertz BCL pulse. a** Schematic of BCL trajectories with different frequency ratio $\omega_L : \omega_R$ and amplitude ratio $I_L : I_R$. (**a1**) A trefoil pattern by $\omega_L : \omega_R = 2 : 1$ and $I_L : I_R = 1 : 1$, (**a2**) a triangle pattern by $\omega_L : \omega_R = 2 : 1$ and $I_L : I_R = 1 : 4$, and (**a3**) a $C_5$ symmetric star pattern by $\omega_L : \omega_R = 3 : 2$ and $I_L : I_R = 1 : 4$. **b** Energy diagrams of the DFG process in circular polarization bases. (**b1**) LCP DFG for frequency $2\omega$ and (**b2**) RCP DFG for frequency $\omega$ are shown as examples. **c** Schematic of the experimental setup for BCL pulse generation. **d** Spectrogram of the designed NIR pulse for generating the trefoil-pattern BCL pulse. The red spectrum represents the measured NIR spectrum after the pulse shaper. Three bands with controlled helicity are used for the generation of the BCL pulse: *U*-band (upper), *C*-band (central), and *L*-band (lower).

of $\omega_L + \omega_R$) in the ideal scenario. The polarization-shaped NIR pulse is directed onto a 10 μm-thick GaSe crystal for DFG. The electric field waveforms of the generated multi-terahertz pulses are characterized as vector values at each delay time by the polarization-modulated electro-optic sampling method (POMEOS)[42].

The electric field of a BCL can be expressed as

$$\boldsymbol{E}(t) = \text{Re}\left(E_L(t)\boldsymbol{e}_L e^{i(\omega_L t + \alpha)} + E_R(t)\boldsymbol{e}_R e^{i\omega_R t}\right), \quad (2)$$

where $\boldsymbol{e}_L = (1, -i)/\sqrt{2}$ and $\boldsymbol{e}_R = (1, i)/\sqrt{2}$ are the basis vectors of the LCP and RCP, $E_L$ and $E_R$ are the amplitudes, and $\alpha$ is the relative phase between the two colors. The diverse trajectories of the BCL field can be achieved by adjusting the frequency ratio $\omega_L : \omega_R$, the intensity ratio $I_L : I_R$, and the relative phase $\alpha$. In Fig. 1a.1, a calculated trefoil pattern is shown for $I_L : I_R = 1 : 1$ and $\omega_L : \omega_R = 2 : 1$. Changing $\alpha$ rotates the trajectory in the $xy$-plane by $\omega_R \alpha / (\omega_L + \omega_R)$ (Supplementary Information Note 2). Figure 1a.1 shows the case of changing $\alpha$ by $3\pi/2$ rad, resulting in a rotation of $-\pi/2$ rad. By adjusting the intensity ratio to

$I_L : I_R = 1 : 4$, the trajectory transforms into a triangle-like pattern in Fig. 1a.2 while maintaining the $C_3$ symmetry. Similarly, Fig. 1a.3 depicts the scenario of changing the frequency ratio to $\omega_L : \omega_R = 3 : 2$, resulting in a $C_5$-symmetric pentagram pattern. Notably, these degrees of freedom in BCL can be programmatically configured, owing to the computer-controlled SLM in our approach.

## BCL pulse generation and controllability of its parameters

The experimental results for the waveform of a BCL pulse with $\omega_L/2\pi = 34$ THz, $\omega_R/2\pi = 17$ THz, and $I_L : I_R = 1 : 1$ are presented in Fig. 2a.2. The trajectory of this pulse projected onto the $xy$-plane exhibits clear $C_3$ symmetry, consistent with the calculation (Fig. 1a.1). Figure 2a.1 displays the power spectra of LCP and RCP components. The peaks centred at 34 THz in the LCP component and at 17 THz in the RCP component serve as clear evidence of the superposition of harmonic frequencies. Thus, we successfully validated the principle of generating desired multi-terahertz BCL pulses using a polarization-tailored NIR pulse.

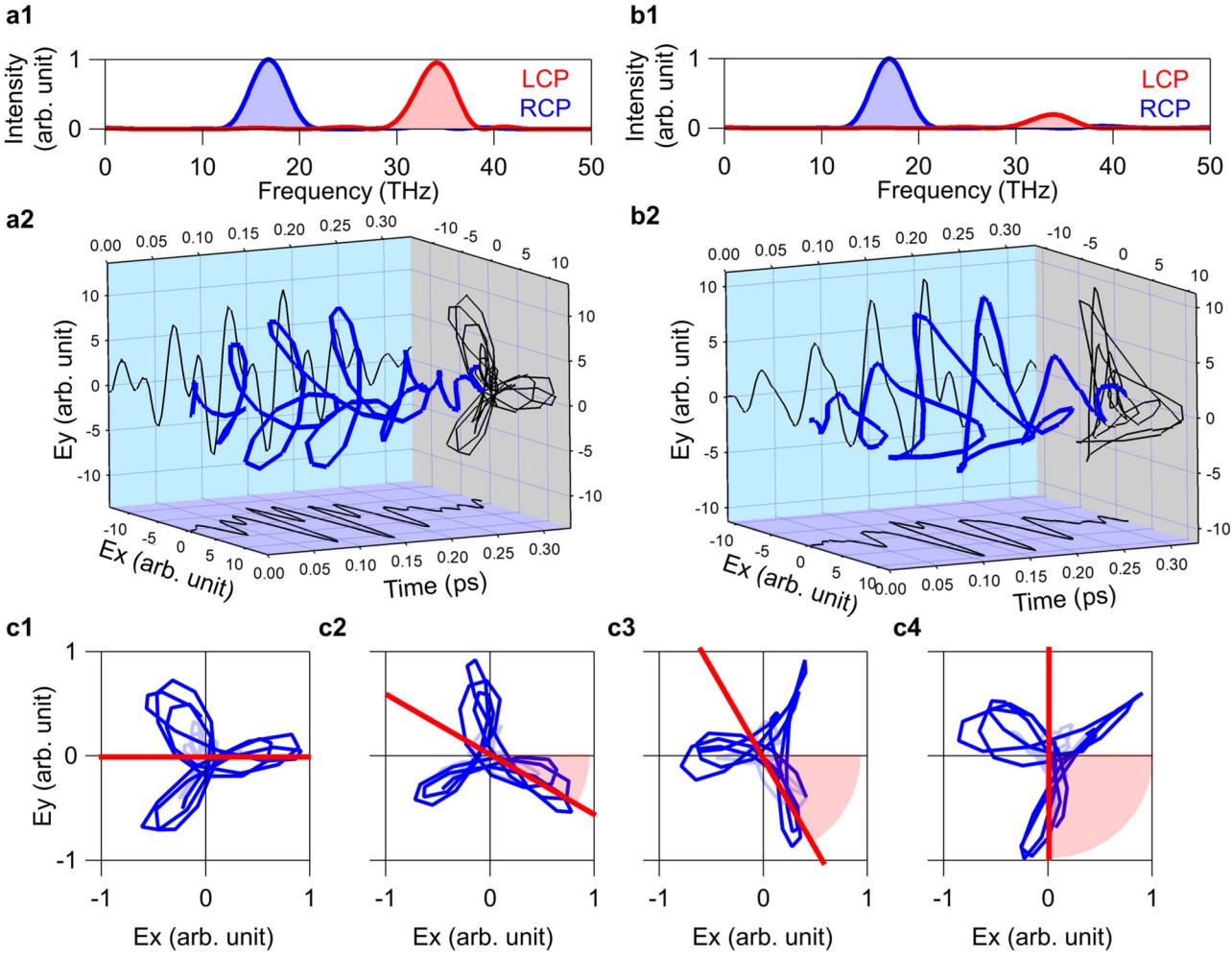

**Fig. 2 | Control of shape and orientation of BCL pulse with $C_3$ symmetry. a** Multi-terahertz BCL pulse with $C_3$ symmetry composed of two colors centered at 17 THz and 34 THz. (**a1**) Power spectra of LCP and RCP components. (**a2**) Three-dimensional plot of the field trajectory. **b** Corresponding results for different intensity ratio ($I_L/I_R = 0.20$) of the two colors from (**a**) ($I_L/I_R = 1.0$). **c** Rotation angle variation of the trajectory in the $xy$-plane. The data obtained along the original direction (**c1**) corresponds to the BCL pulse of (**a**). The rotation angle is increased by 30° per step in **c2**–**c4**. The red dotted line shows the orientation angles of 0°, 30°, 60°, and 90° which we set in the SLM on software and forms the angle indicated by the red shading.

In the following, we demonstrate the controllability of the shape, orientation, rotational symmetry, and helicity of the light field vector trajectory. First, by tuning the ellipticity of the $U$-band, we varied the amplitude ratio between the two colors to control the shape of trajectory. The DFG efficiency for $E_L$ is maximized for an RCP ($\tan\eta = -1$) $U$-band and can be continuously decreased to almost zero by approaching $\tan\eta = 1$. Figure 2b.1 illustrates the result of BCL pulses with intensity ratios $I_L/I_R$ of 0.20, obtained by setting the $U$-band's ellipticity to 0.51. The trajectory of the field in the $xy$-plane changed from a trefoil to a triangle pattern, aligning with the theoretical calculation (Fig. 1a.2). Strictly speaking, a $U$-band with an ellipticity different from that of the $L$-band allows unwanted DFG between these bands. Nevertheless, the NIR intensities in both bands were maintained at considerably lower levels than the $C$-band; Thus, the DFG efficiency was negligibly small.

Second, we realised the orientation control of the BCL trajectory as shown in Fig. 2c. The BCL pulse in our scheme is expressed as follows:

$$\widetilde{E}_L^{\text{THz}}(\omega) \propto \int_{\Omega_0-\Delta\Omega/2}^{\Omega_0+\Delta\Omega/2} d\Omega'\, \widetilde{E}_R^U(\Omega'+\omega)\left[\widetilde{E}_L^C(\Omega')\right]^*, \quad (3-1)$$

$$\widetilde{E}_R^{\text{THz}}(\omega) \propto \int_{\Omega_0-\Delta\Omega/2}^{\Omega_0+\Delta\Omega/2} d\Omega'\, \widetilde{E}_L^C(\Omega')\left[\widetilde{E}_R^L(\Omega'-\omega)\right]^*, \quad (3-2)$$

where $\Delta\Omega$ represents the bandwidth of the NIR bands, the subscript denotes LCP (L) or RCP (R), and the superscript denotes the band ($U$, $C$, or $L$) in the NIR or multi-terahertz frequency (THz). The presented equations indicate that the phase in the multi-terahertz frequency region can be manipulated by applying an offset phase in the NIR pulse. The relative phase $\alpha$ in Eq. (2) corresponds to the phase difference between $\widetilde{E}_L^{\text{THz}}(\omega_L)$ and $\widetilde{E}_R^{\text{THz}}(\omega_R)$ and can be shifted by adding an offset phase $\alpha$ to the $U$-band using the pulse shaper. This control results in the rotation of the orientation of the BCL trajectory by $\omega_R\alpha/(\omega_R+\omega_L)$. Figure 2c.2–4 depict experimental results successfully rotating the BCL trajectories clockwise by 30°, 60°, and 90°, respectively. The angles determined by the relative phase given in the SLM is indicated by red dashed lines in Fig. 2c.1–4, exhibiting that the orientation can be accurately controlled with respect to the set values. This orientation control is possible for all the trajectories in Fig. 1a.1–3 (Supplementary Information Note 3).

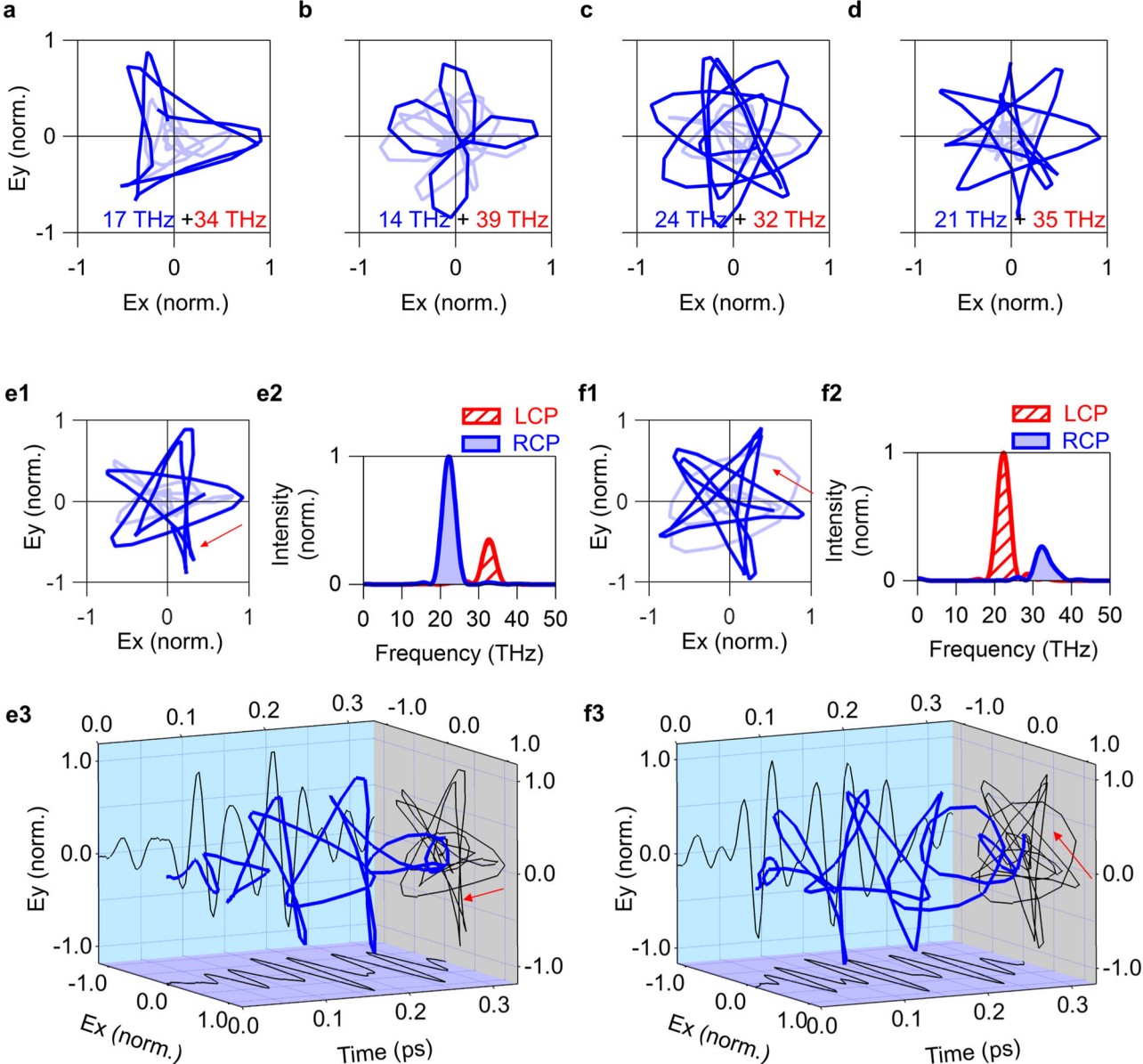

**Fig. 3 | Control of rotational symmetry and helicity of BCL pulses. a–d** Control of rotational symmetry by changing the frequency ratio of the two colors in the BCL pulse. Trajectory in the $xy$-plane of a multi-terahertz pulse with (**a**) $C_3$, (**b**) $C_4$, (**c**) $C_7$, and (**d**) $C_8$ at each frequency ratio. The lower right numbers in the figures indicate the central frequencies of the LCP (red) and RCP (blue) components. **e** Multi-terahertz BCL pulse with $C_5$ symmetry composed of two colors centered at

22 THz and 33 THz. (**e1**) Trajectory in the $xy$-plane, (**e2**) power spectra in circular polarization bases, and (**e3**) three-dimensional plot. The trajectory rotates clockwise as indicated by the red arrow in **e1**. **f** Corresponding results for the opposite helicity compared with (**e**). The trajectory rotates counterclockwise as indicated by the red arrow in **f1**.

Third, the order $n$ of the $C_n$ rotational symmetry was changed, as shown in Fig. 3a–d and 3e.1 for $n$ = 3, 4, 7, 8, and 5, respectively. By tuning the frequency of the three bands ($U$, $C$, and $L$), the DFG frequency in the multi-terahertz region can be controlled within the range 14–39 THz. The dark blue colors in Fig. 3a–d and 3e.1 emphasize the central parts of the pulse envelopes, where the rotational symmetries of BCL pulses were successfully controlled. The $C_3$ and $C_5$ symmetries were clearly realised over a few cycles in Fig. 3a and 3e.1. In contrast, the $C_4$-like pattern in Fig. 3b was achieved only in a limited time window, and the trajectory was considerably distorted away from the pulse center. This result can be ascribed to a slight deviation of $n_1\omega/2\pi = 14$ THz and $n_2\omega/2\pi = 39$ THz from the ideal ratio $n_1 : n_2 = 1 : 3$. To realize the $C_n$ pulse ($n$ = 3, 4, 5, 7, and 8), the required frequency ratio is $n_1 : n_2 = 1 : 2, 1 : 3, 2 : 3, 3 : 4$, and $3 : 5$ respectively. Among them, generating the $C_4$ BCL pulse requires a

pair of largely separated frequency components $n_1\omega$ and $n_2\omega$ by a factor of 3, which must be selected within the available bandwidth. This is also the reason why the $C_6$ pulse was not demonstrated in this work because it requires $n_1 : n_2 = 1 : 5$, i.e., the factor of 5 between $n_1\omega$ and $n_2\omega$. For a large number of $n(>7)$, on the other hand, the paired frequencies are close to each other so that they can be easily chosen in the bandwidth available. However, it requires many oscillation cycles to complete the full trajectory with $C_n$ symmetry. Therefore, generation of the well-designed $C_{n>7}$ BCL pulse requires a certain degree of monochromaticity for each color. By changing the spectral resolution of phase control in the $4f$ system, the monochromaticity could be improved, but there is a trade-off with a decrease in the light intensity. More information is provided in Supplementary Information Note 4. While some problems remain for $n$ = 4 and $n$ > 7, the $C_3$ and $C_5$ BCL pulses were successfully generated,

which covers most of the light-field control experiments in solids theoretically anticipated[26-31].

Fourth, Fig. 3e–f demonstrate the control of BCL pulse helicity, i.e., the direction of the spiral trajectory in the xy-plane. The helicity is reversed by inverting the sign of ellipticity assigned to U-, C-, and L-bands. Figure 3e.1–3 display a multi-terahertz $C_5$ BCL pulse with a clockwise trajectory in the xy-plane, defined as positive helicity. Figure 3f.1–3 present its counterpart with negative helicity. Figure 3e.1 and 3f.1 depict the trajectory of the BCL pulse in the xy-plane, with the red arrow indicating the direction of time passage. The reversal of the rotation direction is also evident in the inversion of LCP and RCP components in the spectrum, as shown in Fig. 3e.2 and 3f.2.

## Discussion

Finally, we address the relative phase stability between two colors in BCL pulses, as the relative phase shift induces rotation of the BCL pulse trajectory. In our approach, the multi-terahertz pulse is generated by intra-pulse DFG from an NIR pulse propagating in a single beam path except for the 4f-setup, ensuring robustness of the relative phase against disturbances. Additionally, our NIR source, derived from a Yb laser with highly stable output and solid-based pulse compression[41], offers advantages in generating phase-stable multi-terahertz pulses[41]. We evaluated two fluctuation components: (i) jitter of the entire pulse along the t-axis, and (ii) a fluctuation in the relative phase between the two colors, which manifests itself as rotation of the field trajectory in the xy-plane. In our experiments, (i) the standard deviation of the electric-field-peak delay time was 0.23 fs, and (ii) the standard deviation of the azimuthal angle of the field at the peak was 14.7 mrad (0.84 deg) over 1 h. Note that this is a passive stability inherent to this setup, and the stability could be further improved if the relative phase α is monitored and actively adjusted on software. Our result demonstrates excellent stability of the temporal waveform as well as of the in-plane trajectory, both beneficial for data accumulation. The stability of trajectory orientation is crucial for the BCL experiment in solid-state systems such as Floquet engineering and valleytronics[26,30,31], in contrast to previous applications of BCL to HHG experiments in gaseous systems where atoms or molecules are randomly oriented. The fluctuation of orientation in our system (14.7 mrad, 0.84 deg) corresponds to the beam path lengths difference of 60 nm that was achieved without mechanical feedback owing to the intra-pulse DFG method.

We would like to emphasize the convenient and broad tunability of BCL frequencies in our method, spanning almost two octaves without any waveplate or other optical element in the mid-infrared or multi-terahertz range, which would find versatile applications in various kinds of target materials. The maximum field amplitude of BCL pulses is currently estimated as 100 kV/cm for an NIR pulse energy of 50 μJ (Methods). This amplitude could be further increased to >1 MV/cm by using a stronger NIR pulse and/or a more efficient pulse-compression method, such as a hollow-core fiber[43,44]. Our results pave the way for new avenues in lightwave control of spatial inversion symmetry and rotational symmetry of materials in Floquet engineering and valleytronics.

**Note added in proof:** After submission of this paper, Tyulnev et al.[45] and Mitra et al.[46] reported on experiments using $C_3$-symmetric BCL for valleytronics with much higher frequencies (>100 THz) up to the NIR range.

## Methods

### Multi-plate broadening and pulse compression

A Yb:KGW regenerative amplifier (PH-2-2mJ-SP, Light Conversion) with a repetition rate of 3 kHz, pulse width of 160 fs, and central wavelength of 1030 nm was employed as a light source. The output exhibited exceptional stability, with shot-by-shot pulse energy fluctuation <0.04% in standard deviation at a high repetition rate. 30% of the output (0.6 mJ) was utilized for the multi-plate broadening method[40,41].

Broadening and compression were conducted in two stages, resulting in a pulse width of ~12 fs. Short-term shot-by-shot pulse fluctuation was <0.05%. To obtain good stability, we applied active feedback to a mirror mount for locking the input beam position to the multi-plate setup. The output from the multi-plate broadening is 55 μJ. This NIR compressed pulse is split into the waveform shaping (50 μJ, 90% of the output) and the electro-optic (EO) sampling light (5.5 μJ, 10% of the output).

### Optical pulse shaper

The pulse shaper comprised a computer-controlled SLM (SLM-S640d, JENOPTIK) situated on the Fourier plane of a 4f configuration, whose optical path length is 60 cm. The SLM featured two phase masks with 640 pixels, each independently operating on the phase of linearly polarized components at ±45° from horizontal[47,48]. The incident horizontally polarized laser pulse, compressed to ~12 fs, was diffracted by transmission diffraction grating (T-1000-1040-31.8 × 12.3–94, Light-Smyth), with a high diffraction efficiency for both p- and s-polarization in the region spanning 800 nm to 1200 nm. By using a concave mirror (focal length of 150 mm), each frequency component was focused onto the liquid crystal pixels in the SLM placed on the Fourier plane. The phase masks controlled spectral phases $\phi_A(\omega)$ and $\phi_B(\omega)$ for linear polarization in the 45° and −45° directions, respectively. In the latter half of the 4f-geometry, different frequency components were recombined to form the shaped laser pulse. The 4f system was all covered by a box made of black polypropylene, which suppressed the temperature change to <0.1 °C. Any additional active feedback is not necessary in the 4f system to obtain the phase stability shown in this work.

### Control of intensity ratio and orientation

To change the intensity ratio, the ellipticity tan η of the NIR bands was manipulated. For example, in Fig. 2a, ellipticities of −0.51 and −1.0 were assigned to the L-band and U-band, resulting in a controlled intensity ratio $I_L/I_R$ of 1.0. As shown in Fig. 2b, we assigned ellipticities of −1.0 to the L-band and 0.51 to the U-band, resulting in a controlled intensity ratio $I_L/I_R$ of 0.20. For orientation control, an offset phase φ was added as follows: offset phases of 0.2π, 0.7π, 1.2π, and 1.7π corresponded to the orientation of 0°, 30°, 60°, and 90° in Fig. 1c.1–4, respectively.

### Multi-THz BCL pulse generation

Intra-pulse DFG using GaSe crystals generated multi-terahertz pulses. GaSe crystals with a thickness of 10 μm in the ab-plane were used for broadband DFG in the 14–40 THz range. We used a crystal with $C_3$ symmetry, which can transfer the angular momentum of ±3ℏ to the lattice at the present experimental condition. The higher-frequency range of generated multi-terahertz pulses is limited by the bandwidth of NIR pulse shown in Fig. 1d. Furthermore, the higher frequency side is also limited by the absorption by water vapor in air significantly beyond 40 THz. It could be avoided by purging with nitrogen gas, but we did not use it in this work because the flow of the air would lower the stability of the optical system. The lower-frequency range is limited by phonon absorption in GaSe crystal.

### Angular momentum conservation law

The conservation law of angular momentum using $C_3$-symmetric materials has been discussed for the case of second harmonic generation in 1960's[38] and stimulated Raman scattering of magnon excitation[39]. A similar conservation law of angular momentum can be applied to any nonlinear optical processes, where angular momentum of ±nℏ around the optical axis can be dumped on the $C_n$-symmetric environment. Since the DFG is a second-order process, the pure angular momentum transferred from two incident photons is 0 or ±2ℏ. On the other hand, a DFG photon requires an angular momentum

of $\pm\hbar$. Therefore, an excess angular momentum of $\pm\hbar$ or $\pm 3\hbar$ should be transferred to the crystal, which means that DFG is allowed only for $n=1$ or 3. In the case of $n=1$, the selectivity of the helicity disappears because $+\hbar \equiv -\hbar(\mathrm{mod}\hbar)$. Therefore, $n=3$ is the only solution for selectively generating $\pm\hbar$ with DFG process. GaSe crystal that we used in this work satisfies this requirement.

## Detection of the multi-terahertz BCL pulse

After the generation of multi-terahertz light, the near-infrared light was cut by a 0.5 mm-thick Ge filter. Multi-terahertz pulses were measured by polarization-modulated electro-optic sampling (POMEOS)[42] using NIR pulses with a width of ~12 fs as sampling light. The NIR pulse was characterized using a second-harmonic generation-based frequency-resolved optical gating (SHG-FROG) method (Supplementary Information Note 5). The polarization was modulated by a photoelastic modulator (PEM), and the crystal used for detection was a GaSe crystal with a thickness of 10 μm.

## Evaluation of phase stability

The $C_5$ pulse waveform was measured repeatedly for 1 h to evaluate stability, primarily affected by two fluctuations: (i) jitter of the entire pulse on the $t$-axis and (ii) fluctuation in the relative phase between the two colors in the BCL pulse. (i) The jitter was evaluated by analyzing the time delay at which the absolute value of the electric field $E_{\mathrm{abs}} = \sqrt{E_x^2 + E_y^2}$ was the maximum. (ii) To evaluate the relative phase, we analyzed the azimuthal angle at the peak delay time because the change in the relative phase manifests itself as rotation of the BCL pulse. To isolate the relative phase from the time-delay jitter, the envelope center of gravity of the RCP components was measured as the instantaneous polarization angle at the envelope peak time. According to the observations, (i) the standard deviation of the electric-field-peak delay time was 0.23 fs and (ii) the standard deviation of the azimuthal angle of the field at the envelope peak was 14.7 mrad (0.84 deg) for 1 h, corresponding to the fluctuations in the $t$-axis and $xy$ plane, respectively. Further details in Supplementary Information Note 6.

## Consideration for conventional methods using quarter waveplates

In the visible or NIR region, the $C_3$ BCL pulses can be generated by using fundamental light ($\omega$) and its second harmonics ($2\omega$), with making them circularly polarized beams independently in separated optical paths in the Mach-Zehnder-like geometry[23] or with achromatic quarter-half waveplate in the in-line geometry[37]. In particular the latter is favorable for the excellent phase stability. To apply these methods in the multi-terahertz frequency range, however, dispersion-less transparent optical elements are required, whereas most of the existing materials show the phonon absorptions in this range. For a limited frequency window, there are commercially available quarter waveplates in the multi-terahertz frequency range, but they are expensive because of the problems of toxicity, low mechanical strength, and chemical degradation[35]. In addition, it must be replaced with another waveplate if one wants to flexibly change the wavelength in BCL. Separation of the optical beam paths for each color to insert the quarter waveplate would also lower the stability of relative phase.

## Data availability

All data needed to evaluate the conclusions in the paper are present in the paper and/or the Supplementary Information. Additional data and code related to this study are available from the corresponding authors upon request.

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

## Acknowledgements

This work was supported by JST PRESTO (Grants No. JPMJPR2006 and No. JPMJPR20LA) and by JST CREST (Grant No. JPMJCR20R4). R. M. also acknowledges partial support by attosecond lasers for next frontiers in science and technology (ATTO) in the Quantum Leap Flagship Program (MEXT Q-LEAP).

## Author contributions

N.K. conceived this project. K.O. and N.K. developed the system and performed the experiment with the help of Y.M. and R.M. K.O. analyzed the data with the help of N.K. and Y.M. K.O., N.K., and R.M. wrote the manuscript with substantial feedback from Y.M.

## Competing interests

The authors declare no competing interests.
