## [Peer Review File · Nature Communications]

REVIEWER COMMENTS

Reviewer #1 (Remarks to the Author):

Please see the attached Reviewer report.

Reviewer #2 (Remarks to the Author):

The paper is entitled "Programmable generation of counterrotating bicircular light pulses in the multi-terahertz frequency range" and it explores the generation of counterrotating bicircular light pulses in the mid-infrared or multi-terahertz region. The paper introduces a methodology for generating phase stable counterrotating bicircular light within the 10-40 frequency band. This methodology is achieved with an slm and intra-pulse difference frequency generation.

The results presented in the paper are certainly intriguing and hold potential significance for the community interested in ultrafast science and condensed matter physics. However while the results are certainly intriguing I have reservations regarding the novelty of the paper. In particular polarization control of bicircular light has been demonstrated previously. In particular in the article (<https://doi.org/10.1364/JOSAB.456066>). However the following article is also very relevant to the paper. (<https://doi.org/10.3390/photonics10070803>). It is noteworthy that both of these articles have been omitted by the authors. Without the novelty claim on the polarisation control of bicircular light, I do not believe that being programmable and not requiring waveplates or polarizers is a sufficient claim for publication within nature communications.

Additionally, i have some minor concerns

- 1). Describing something in S^{-1} is highly unusual, especially when you frequently refer to it as terahertz, why not just use Hz?
- 2). In my opinion "intra-pulse difference frequency generation (DFG)" should be modified to "optical rectification" (a well studied phenomena under this name) to better align to the nomenclature used by the terahertz community.

Reviewer report for manuscript ID: NCOMMS-24-04948

Title: Programmable generation of counterrotating bicircular light pulses in the multi-terahertz frequency range

Authors: Kotaro Ogawa, Natsuki Kanda, Yuta Murotani, and Ryusuke Matsunaga

In this paper, the authors demonstrate the generation of counter-rotating bicircular light fields in the frequency range between 10 and 40 THz (central wavelength from 7.5 to 30 μm). Controlling the polarization state of light pulses in the mid-infrared (MIR) spectral region is particularly challenging. The key idea of the manuscript is to tailor the polarization state of a strong near-infrared (NIR) pulse and use it for generating counter-rotating bicircular MIR radiation via intra-pulse difference frequency generation (DFG). Control over the polarization state of the distinct NIR wavelengths is achieved by a computer-controlled spatial light modulator (SLM).

This frequency range in the mid-infrared spectral region is of particular interest for the investigation of electron dynamics in solids, as the associated photon energies (165 to 40 meV) are comparable with the typical energy scales governing electronic correlations in solid-state materials^[R1]. In addition, precisely driving electron dynamics with customized light transients, represents a crucial preliminary step towards the development of lightwave electronics^[R2]. In my opinion, the manuscript represents a significant advancement in the field of ultrafast optics, with possible relevant applications in condensed matter physics on ultrashort time scales.

Still, it presents some major open points which require further clarifications by the authors. The “Discussion” section needs to be significantly expanded and clarified, in particular addressing the meaning of the pulse jitter and how it is experimentally evaluated. Moreover, the authors do not discuss possible limitations and future perspectives of their work. In conclusion, I consider this work to be suitable for publication in Nature Communications after the following concerns have been thoroughly addressed.

Major concerns

- Lines 71-72: the authors affirm that controlling the polarization of multi-THz pulses is difficult for the lack of dispersion-free optical elements. This statement is crucial, as it is the reason why their work is relevant. However, they do not include any citations or further comparison with the published literature to support it. Better framing their work, highlighting previous attempts (if any) and their shortcomings, or discussing possible alternative approaches, would better highlight the relevance of their work.
- Line 84: the generated spectrum spans from 10 to 40 THz. What prevents them from extending this frequency range? Is it related with the nonlinear crystal chosen? If not, where does it come from? Are there more intrinsic limitations? What could be the ultimate limit?
- Lines 102-104: linked to the previous comment, they highlight the C_3 symmetry of GaSe, which allows satisfying both energy and momentum conservation in the DFG process. Would the same scheme and conservation law apply for crystals with different symmetry properties? And eventually considering also orbital angular momentum of light? Briefly commenting on this topic would give generality to their discussion, highlight its relevance, and suggest possible future experimental developments.
- Line 141-142: to prevent DFG from undesired frequencies, the authors transform them to right circular polarization and introduce a 400-fs group delay. If I understand properly, transforming them to the same circular polarization prevents DFG because it will not be possible to satisfy the angular momentum conservation rule. Then, why are they further delayed by 400 fs? Are they still interacting with radiation in some of the other bands? To demonstrate that the delayed components are not giving any interaction, that could be detrimental for pulse contrast, the authors should check the generated counter-rotating bicircular light transients even for time values longer than 400 fs.
- Line 155-156: the relation linking the additional phase α with the rotation of the field transient in the xy-plane is not demonstrated. If the authors have derived it for the first time, they should add the derivation in the methods or in the supplementary materials. If not, a citation should be added.
- Lines 182-185: the authors control the amplitude of the left circularly polarized MIR radiation by reducing the ellipticity in the U band. As they explain, a possible drawback of this approach is that there could potentially be DFG between the U- and L-bands, as now they have different polarization. Why is it necessary to change the polarization of the U-band? Would not have it been enough to reduce its intensity with the SLM, without changing its polarization state?

- Lines 201-206 and Fig. 3a-d: the authors demonstrate that changing the ratio between ω_L and ω_R the rotational symmetry changes. However, they also demonstrated in Fig. 2 that also the amplitude ratio between the left and right circularly polarized components is fundamental for shaping the light transients. It is thus necessary to add this information to the discussion.
- Lines 227-229: the authors present numerical values supporting the stability of the setup, without showing the related data. A few additional details are provided in the methods section “Evaluation of the phase stability”. In my opinion, this section should be significantly extended, adding one figure, and including additional experimental details. As an example, they do not specify how many times the time delay of the peak electric field and the azimuthal angles were sampled over one hour, if each data point was the result of averaging several POMEOS measurements, and the number of laser shots required for each POMEOS measurement. Can the authors comment on the shot-to-shot stability of the proposed scheme?
- Lines 243-244: how is this value estimated? What is the NIR pulse energy associated with such a field? Would it be possible to increase it? What is its main limitation?
- Lines 292-304 and 218-238: the meaning of the jitter of the entire pulse on the t-axis is unclear. How do the authors define a meaningful zero of the time axis? If I understand properly, they retrieve the MIR transient using the POMEOS technique. From Ref. [26] of the manuscript, the time-axis in this two-color measurement corresponds to the delay between the MIR field and a gate pulse. As in any pump-probe experiment, this is not an absolute time axis: shifting arbitrarily both pulses in time, the POMEOS measurement will not change, as it only depends on the gate and MIR delay. Therefore, what is the meaning of this jitter? How can the authors attribute it to a shift of the MIR transient, and not to the gate pulse? How can the timing jitter (and the same applies to the BCL rotation) be distinguished from a change in the carrier-envelope offset of the MIR transient?
- Figs. 2c1-4, 3a-d, 3e1 and 3f1: the dark blue lines hide some light blue lines. Why are these lines plotted in different colors? Do they have a different meaning? If they are field values where the symmetry of the BCL is not particularly evident, the authors should critically discuss them, explain where they come from and quantify them (e.g., comparing their maximum field value with the peak field amplitude) in the “Discussion” section. In addition, in the E_x - E_y plots, using a color scheme where different sections of the line are represented with different colors encoding distinct time values, and using the same color scheme for 3D graphs (Figs. 2a2, 2b2, 3e1, and 3f3) would help understanding how the electric field of the BCL evolves in time.
- General comment: in my opinion, the manuscript lacks a thorough discussion of the parameters of the generated BCL pulses (energy, duration, pulse contrast, ...), what currently limits them, what is the main limitation to achieving full polarization control over the MIR transient (if there is any), and how to possibly overcome these limitations in the future. I think this information to be crucial for future experimental applications of their approach.

Minor comments

- Line 34: the authors cite three exemplary experiments, two from the dawn of attosecond science in solids (Refs. [1] and [2] of the manuscript) and one demonstrating Bloch oscillations with HHG in solids (Ref. [3]). Despite their importance, several additional groups have investigated similar phenomena in distinct experimental conditions. To better picture also recent developments on these topics, and being this an introductory paragraph, I would suggest also citing some recent reviews on these topics (e.g., Refs. [R2]-[R4]).
- As a general comment on figures, indicating panels with a letter and sub-panels again with a number (e.g., Figs. 1a1 and 1a2 on line 46) is unusual. My suggestion is to separate the letter from the subsequent number with a dot (e.g., Figs. 1a.1 and 1a.2) for clarity.
- Lines 65-66, point (i): in this form, this phrase is not always true. As an example, electron-electron scattering can be extremely fast^[R5], even reaching the sub-femtosecond time scale. Electron-phonon scattering, instead, typically takes place on longer time scales. My suggestion is to rephrase it as: “(i) it oscillates faster than typical electron-phonon scattering times [...]”.
- Lines 68-69, point (iii): if I understood the authors’ point properly, they wanted to say that, for the same peak electric field amplitude, multi-THz light fields lead to a larger amplitude of the vector potential compared to visible light. This comes from the inverse scaling of the vector potential amplitude with the frequency. However, increasing the pulse energy or tighter focusing for visible

- pulses could lead to the same vector potential amplitude. If the authors want to keep this point as an advantage of multi-THz fields, it must be clarified.
- Lines 69-70, point (iv): this point is ambiguous. In a two-band system, inter-band excitation can be categorized by using two adiabaticity parameters^[R6]: the ratio between the energy bandgap and the photon energy and the Keldysh parameter. Depending on the experimental conditions, the interaction of light pulses can or cannot lead to a residual excited electron population^[R7] and, eventually, damage the material. Thus, at least four quantities must be considered (the electron-hole effective mass, the energy gap, the photon energy, and the peak field amplitude), and two of them depend on the sample considered. Thus, this point must be either clarified, or removed from the discussion.
 - Lines 96-104: the authors discuss the energy and angular momentum conservation that is crucial for their scheme. For light pulses, they only relate the angular momentum to the polarization state of light. I would thus suggest replacing “angular momentum” with “(spin) angular momentum” when it is referred to the properties of the light beam.
 - Lines 125-126: why is the phase modulated along the azimuthal angles of $\pm 45^\circ$? Is there some physical reason for this? Is it connected with the implementation of the SLM?
 - Line 179: while in the discussion of Fig. 1 the authors refer to the amplitude of the ω and 2ω light fields, here and in Fig. 2 they present an intensity ratio. Even though the two quantities are clearly connected, probably choosing only one for the whole manuscript would make it clearer. In addition, an intensity ratio of 0.2 gives an amplitude ratio of $\sqrt{0.2} \approx 0.45$. Repeating the simulation of Fig. 1a2 with the same ratio as the experiment would allow a more direct comparison.
 - Line 187: the authors discuss the orientation control of the BCL trajectory. If this type of control is general and valid for all the trajectories in Fig. 1a1, 1a2, and 1a3, the authors should clearly state it and demonstrate it with supplementary data. If instead it is specific of the trajectory in Fig. 1a1, they should clearly state it.
 - Line 196: applying the offset phase α to the L-band would have given the same result? Would the formula on line 198 remain the same, but switching R with L and vice versa? If this is the case, then in the example of Fig. 1a1 $\omega_L = 2\omega$ and $\omega_R = \omega$, so for the same α the rotation should be larger.
 - Line 221-222: as correctly highlighted, in the $4f$ setup different components follow different optical paths. Have the authors quantified the possible timing jitter between the different (U, C, L) bands? Does the setup require any active or passive stabilization scheme?
 - Line 238: a beam path difference of 60 nm corresponds to a timing jitter of ~ 200 as. In attosecond science, this mechanical stability is hard to achieve without any mechanical feedback (see for example blue line in Fig. 5a from [R8], where the phase error in the open loop condition corresponds to a phase delay shift of several femtoseconds over less than one hour). Can the authors comment one on how they achieved such an impressive stability? What is the length of the $4f$ setup?
 - Line 258: a more quantitative value on the pulse duration is required. What is its exact value? Is it expressed as the intensity full width at half-maximum duration? How is it characterized? What is the associated Fourier transform limited duration?
 - Lines 261-263: including a reference to published literature describing the basic working principle of the SLM in the $4f$ configuration would expand the readership of the manuscript.
 - Line 268: is the concave mirror a spherical mirror? What is the focal length?
 - Line 287: are the MIR and NIR beams impinging orthogonally on the Ge filter? What is its thickness?
 - Despite making use of a stretchable hollow-core fiber, the main point of Ref. [27] is the generation of soft x-rays in the water window via high-order harmonic generation (HHG). Ref. [R9], being a review on the generation of high-energy few-cycle laser pulses, is probably better.
 - Figs. 2c1-4: how is the red dashed line obtained? If it is just a guide to the eyes, it should be stated in the caption.
 - Figs. 3a-d: strictly speaking, the frequency ratio in the bottom right corner (e.g., 17:34 in Fig. 3a) is a dimensionless quantity. I suggest to not express this quantity as a ratio.

Typos and suggestions

- Lines 42-43: trajectories of the light field vector \rightarrow trajectories of the light electric field vector.

- Lines 44-45, “Counterrotating [...] delineate”: this phrase is difficult to read. I suggest rewriting it as: “Counterrotating bicircular light fields (BCL) with frequencies ω (fundamental) and 2ω (second harmonic) delineate [...]”.
- Line 136: the centre frequency of the C-band \rightarrow the central frequency of the C-band.
- Line 201: Third, order n \rightarrow Third, the order n .
- Line 208-209: the rotating direction of the trajectory \rightarrow the direction of rotation of the trajectory.
- Line 256: Of the output, 11% (0.21 mJ) \rightarrow 11% of the output (0.21 mJ).
- Dashes are missing in the titles of refs. [3] and [13].
- Figs. 2a1, 2b1, 3e2 and 3f2: using a different line type (e.g., dashed) for LCP or RCP would make the difference more evident even in printed black and white versions of the manuscript.
- Figs. 2a2, 2b2, 2c1-4, 3a-d, 3e1, 3e3, 3f1 and 3f3: I suggest normalizing the field amplitude in all panels (e.g., to the maximum field amplitude in the transient). Moreover, panels showing the same quantity, as 3e1 and the right-hand side of 3e3, should have the same normalization.

References

- [R1] Freudenstein, J. et al. Attosecond clocking of correlations between Bloch electrons. *Nature* 610, 290–295 (2022).
- [R2] Borsch, M., Meierhofer, M., Huber, R. & Kira, M. Lightwave electronics in condensed matter. *Nat. Rev. Mater.* 8, 668–687 (2023).
- [R3] Bao, C., Tang, P., Sun, D. & Zhou, S. Light-induced emergent phenomena in 2D materials and topological materials. *Nat. Rev. Phys.* 4, 33-48 (2022).
- [R4] Yang, C.-J., Li, J., Fiebig, M. & Pal, S. Terahertz control of many-body dynamics in quantum materials. *Nat. Rev. Mater.* 8, 518-532 (2023).
- [R5] Chen, C. et al. Distinguishing attosecond electron–electron scattering and screening in transition metals. *Proc. Natl. Acad. Sci. U.S.A.* 114, E5300-E5307 (2017).
- [R6] Heide, C., Boolakee, T., Higuchi, T. & Hommelhoff, P. Adiabaticity parameters for the categorization of light-matter interaction: From weak to strong driving. *Phys. Rev. A* 104, 023103 (2021).
- [R7] Di Palo, N. et al. Attosecond absorption and reflection spectroscopy of solids. *APL Photonics* 9, 020901 (2024).
- [R8] Luttmann, M., Bresteau, D., Hergott, J. F., Tcherbakoff, O. & Ruchon, T. In Situ Sub-50-Attosecond Active Stabilization of the Delay between Infrared and Extreme-Ultraviolet Light Pulses. *Phys. Rev. Appl.* 15, 1 (2021).
- [R9] Nagy, T., Simon, P. & Veisz, L. High-energy few-cycle pulses: post-compression techniques. *Adv. Phys. X* 6, (2021).

List of changes

1. We added a review paper for infrared-transparent materials in the reference [Ref. 35].
2. We added an explanation for the difficulty in using infrared transparent waveplate in the Methods section.
3. We added an explanation for the limitations of available bandwidth in the Methods section.
4. We added a more general argument of the DFG process for other rotational symmetries in the nonlinear crystal in the Methods section.
5. We explained the reason requiring the 400-fs group delay in the Supplementary Information Note 1 and a guide to this in the main text (Line 153).
6. We added a calculation of the relation between the rotational angle of field trajectory and the relative phase in the Supplementary Information Note 2.
7. We added Fig. S3 in the Supplementary Information Note 4 to show the information of relative intensity for each color, and a brief guide to this in the main text (Line 237-240).
8. We added the discussion about relative phase stability of BCL pulses in the Supplementary Information Note 6 and added Fig. S5.
9. We added the required information about peak electric field in the Supplementary Information Note 7.
10. We added explanations for the definition of the jitter in the main text and in the Supplementary Information Note 6.
11. We added an explanation for the different color plots in Figs. 2c. 1-4, 3a-d, 3e. 1 and 3f. 1 and a discussion for the difficulty in generating C_4 and $C_{n>7}$ in the main text (Lines 219-240 and 326-333).
12. We added the information for the parameters of the generated BCL pulses in the Supplementary Information Note 8.
13. We modified the reference list to include some important literature for light field electronics.
14. We separated letters indicating figures from subsequent numbers with dots (e.g., Figs. 1a.1 and 1a.2).
15. We rephrased the expression of electron-phonon scattering (Line 68).
16. We improved the statement in the introduction to discuss the importance of intense multi-terahertz pulses compared with the near-infrared or visible pulses in the main text (Lines 70-73).
17. We modified the explanation for angular momentum in the main text (Lines 104-105).
18. We added a brief explanation and a reference to SLM in the Methods section.
19. We used the intensity ratio in the whole manuscript instead of the field amplitude ratio.
20. We added Fig. S2 to show the controllability of orientation of every trajectory with C_3 or C_5 symmetry in the Supplementary Information Note 3.
21. We explained the case of adding the offset phase to the L -band in the Supplementary Information

Note 2.

22. We explained the evaluation of the timing jitter in the Supplementary Information Note 6.
23. We added the explanation about the feedback of multi-plate broadening in the Methods section.
24. We added the evaluation of compressed pulses in the Supplementary Information Note 5.
25. We added the information of a concave mirror and a focal length in the Methods section.
26. We added the information of the Ge filter in the Methods section.
27. We modified the reference of a hollow-core fiber.
28. We explained the definition of the red line in Figs. 2c, 1-4 in the main text (Lines 210-214) and the figure caption.
29. We modified the expression of the frequency ratios in Fig. 3.
30. We corrected the typos.
31. We corrected the colors used in Figs. 2 and 3.
32. We improved the normalization and plotting of the figures (Fig. 2 and 3).
33. We added a brief comparison with the previous literature using BCL pulses in the visible or near-infrared region (lines 78-85).

Reply to Report of Reviewer #1

Reviewer report and our reply are shown in cyan and black, respectively. The reference numbers below are those updated in the revised manuscript.

In this paper, the authors demonstrate the generation of counter-rotating bicircular light fields in the frequency range between 10 and 40 THz (central wavelength from 7.5 to 30 μm). Controlling the polarization state of light pulses in the mid-infrared (MIR) spectral region is particularly challenging. The key idea of the manuscript is to tailor the polarization state of a strong near-infrared (NIR) pulse and use it for generating counter-rotating bicircular MIR radiation via intra-pulse difference frequency generation (DFG). Control over the polarization state of the distinct NIR wavelengths is achieved by a computer-controlled spatial light modulator (SLM).

This frequency range in the mid-infrared spectral region is of particular interest for the investigation of electron dynamics in solids, as the associated photon energies (165 to 40 meV) are comparable with the typical energy scales governing electronic correlations in solid-state materials [R1]. In addition, precisely driving electron dynamics with customized light transients, represents a crucial preliminary step towards the development of lightwave electronics [R2]. In my opinion, the manuscript represents a significant advancement in the field of ultrafast optics, with possible relevant applications in condensed matter physics on ultrashort time scales.

Still, it presents some major open points which require further clarifications by the authors. The “Discussion” section needs to be significantly expanded and clarified, in particular addressing the meaning of the pulse jitter and how it is experimentally evaluated. Moreover, the authors do not discuss possible limitations and future perspectives of their work. In conclusion, I consider this work to be suitable for publication in Nature Communications after the following concerns have been thoroughly addressed.

We acknowledge Reviewer #1 for esteeming our work and giving many constructive criticisms, with recommending publication after the revisions. We have addressed all the comments below and have made major revisions as follows.

1. Lines 71-72: the authors affirm that controlling the polarization of multi-THz pulses is difficult for the lack of dispersion-free optical elements. This statement is crucial, as it is the reason why their work is relevant. However, they do not include any citations or further comparison with the published literature to support it. Better framing their work, highlighting previous attempts (if any) and their shortcomings, or discussing possible alternative approaches, would better highlight the relevance of their work.

We appreciate Reviewer #1 for the helpful suggestion. Let us explain it in more detail as follows; First of all, the dispersion-less optical elements suitable for our purpose must be transparent in the midinfrared and multi-terahertz frequency range. However, most of the existing materials show the phonon absorptions in this range, which strictly limits the choice of materials. Furthermore, as explained by a review article [Ref. 35], most of the infrared-transparent materials suffer from the problems of toxicity, low mechanical strength, and chemical degradation including deliquescency, which makes difficult and expensive to produce it. For a certain single color, there are commercially available (and expensive) quarter waveplates. But it must be replaced with another waveplate if one wants to flexibly change the wavelength in BCL. Separation of the optical beam paths for each color to insert the quarter waveplate would also significantly lower the stability of relative phase.

In the revised manuscript, we added the review paper as a reference (**Change #1**). In addition, we added explanation for the difficulty of infrared transparent materials in the Methods section (**Change #2**).

• Line 84: the generated spectrum spans from 10 to 40 THz. What prevents them from extending this frequency range? Is it related with the nonlinear crystal chosen? If not, where does it come from? Are there more intrinsic limitations? What could be the ultimate limit?

For the present experimental condition, extending the frequency range is limited by the bandwidth of near infrared pulse shown in Fig. 1d. In addition, absorption by water vapor in air becomes significant beyond 40 THz. Purging with nitrogen gas could avoid the water-vapor absorption, but the flow of the air would lower the stability of optical system. For the lower-frequency side, phonon absorptions in GaSe crystal limits the spectrum.

If we use other crystals, we can cover the different frequency region, e.g. GaP (0.5-7 THz), LiGaS₂ (6.8-16.4 μm , 18-44 THz), and BiBO (1.4-3.1 μm). It would be difficult to cover the entire range with a single crystal. The ultimate limit might reach to NIR or visible if the bandwidth of the supercontinuum is wide enough. However, for such higher frequencies, the use of commercially available optical elements is a reasonable choice to generate the BCL light, as demonstrated in [Ref. 37].

In the revised manuscript, we added this explanation in the Methods section (**Change #3**).

• Lines 102-104: linked to the previous comment, they highlight the C₃ symmetry of GaSe, which allows satisfying both energy and momentum conservation in the DFG process. Would the same scheme and conservation law apply for crystals with different symmetry properties? And eventually

considering also orbital angular momentum of light? Briefly commenting on this topic would give generality to their discussion, highlight its relevance, and suggest possible future experimental developments.

In principle, the same conservation law of angular momentum can be applied to any nonlinear optical processes, where angular momentum of $\pm n\hbar$ can be transferred to the C_n -symmetric system around the optical axis. However, the selectivity of circular polarization in DFG can be achieved only in the C_3 system for the following reason; Because the DFG process is the second-order process, the induced angular momentum generated by two incident photons is 0 or $\pm 2\hbar$. On the other hand, a DFG photon requires an angular momentum of $\pm\hbar$. Therefore, the system should transfer $\pm\hbar$ or $\pm 3\hbar$ to the C_n -symmetric system, which means that DFG is allowed only for $n = 1$ or 3. In the case of $n = 1$, the selectivity of the helicity disappears because $+\hbar \equiv -\hbar \pmod{\hbar}$. Therefore, $n = 3$ is the only solution for selectively generating $\pm\hbar$ with DFG process.

As pointed by Reviewer #1, further consideration including the orbital angular momentum of light is also highly intriguing. Angular momentum conservation of vortex beam has been also discussed in the other nonlinear optical processes such as optical parametric amplification and high harmonic generation. However, we believe that this is far beyond the scope of the present work.

In the revised manuscript, we added the brief explanation in the Methods section (**Change #4**).

- Line 141-142: to prevent DFG from undesired frequencies, the authors transform them to right circular polarization and introduce a 400-fs group delay. If I understand properly, transforming them to the same circular polarization prevents DFG because it will not be possible to satisfy the angular momentum conservation rule. Then, why are they further delayed by 400 fs? Are they still interacting with radiation in some of the other bands? To demonstrate that the delayed components are not giving any interaction, that could be detrimental for pulse contrast, the authors should check the generated counter-rotating bicircular light transients even for time values longer than 400 fs.

As properly expected by Reviewer #1, the group delay for undesired frequencies is indispensable because otherwise they would interact with either of L -, C -, or U - bands and produce undesired DFG. We indeed checked the generated BCL transients for longer time delay to check the effect of the 400-fs-delayed underside part. The added Fig. S1 shows the result of the trajectory with a time window of 1 ps, as shown below. Note that the free induction decay of absorptions in CO_2 and H_2O at around 25 and 40 THz, respectively, and the backside reflection inside the 10 μm -thick electro-optic crystal would also appear in this time region. Compare with the main peak, the signal delayed by 400 fs indicated by the black arrow is sufficiently weak, which demonstrates the validity of our method.

In the revised manuscript, we added Fig. S1 in the Supplementary Information Note 1 with a brief explanation (**Change #5**).

Fig. S1. The BCL trajectory measured with a time window of 1 ps. a, Multi-terahertz BCL pulse trajectories along x -axis (red line) and y -axis (light blue line). **b,** A 3D plot of the field trajectory.

- Line 155-156: the relation linking the additional phase α with the rotation of the field transient in the xy -plane is not demonstrated. If the authors have derived it for the first time, they should add the derivation in the methods or in the supplementary materials. If not, a citation should be added.

We added a derivation of calculation in the Supplementary Information Note 2 (**Change #6**).

- Lines 182-185: the authors control the amplitude of the left circularly polarized MIR radiation by reducing the ellipticity in the U band. As they explain, a possible drawback of this approach is that there could potentially be DFG between the U- and L-bands, as now they have different polarization. Why is it necessary to change the polarization of the U-band? Would not have it been enough to reduce its intensity with the SLM, without changing its polarization state?

The dual SLM used in this work consists of two layers of nematic liquid crystals to modulate the refractive indices in the two orthogonal directions ($\pm 45^\circ$). For a certain purpose of use, the dual SLM could be used to control the light intensity by combining a polarizer. However, in this work, the two degrees of freedom controllable in the SLM are already used to determine the ellipticity $\eta(\Omega)$ and the phase $\phi(\Omega)$, as explained in Eq. (1) in the main text. Therefore, the ellipticity is the only way to control the amplitude of circularly polarized multi-terahertz output.

- Lines 201-206 and Fig. 3a-d: the authors demonstrate that changing the ration between ωL and ωR the rotational symmetry changes. However, they also demonstrated in Fig. 2 that also the amplitude

ratio between the left and right circularly polarized components is fundamental for shaping the light transients. It is thus necessary to add this information to the discussion.

Please see Fig. S3 below, which show the intensity spectra and 3D trajectories corresponding to the controlled rotational symmetries in Fig. 3. As properly expected by Reviewer #1, the intensity ratio was controlled in Fig. 3. For each frequency ratio, the intensity ratio was adjusted to clearly show the rotational symmetry.

In the revised manuscript, we added Fig. S3 in the Supplementary Information Note 4 to show the information of relative intensity for each color, and a brief guide to this in the main text (**Change #7**).

Fig. S3. The intensity spectra and 3D plots of BCL pulses with controlled rotational symmetry. Each trajectory has (a and b) C_3 in Fig. 3a, (c and d) C_4 in Fig. 3b, (e and f) C_7 in Fig. 3c, or (g and h) C_8 in Fig. 3d symmetry.

• Lines 227-229: the authors present numerical values supporting the stability of the setup, without showing the related data. A few additional details are provided in the methods section “Evaluation of the phase stability”. In my opinion, this section should be significantly extended, adding one figure, and including additional experimental details. As an example, they do not specify how many times the time delay of the peak electric field and the azimuthal angles were sampled over one hour, if each data point was the result of averaging several POMEOS measurements, and the number of laser shots required for each POMEOS measurement. Can the authors comment on the shot-to-shot stability of the proposed scheme?

To evaluate the phase stability, we repeatedly measured the BCL electric field trajectories for 1 hour by POMEOS. Each of the trajectories is consisted of 100-point measurements over a 0.33 ps scan range. For each delay point, 900 laser shots (300 ms) are accumulated, and 100-point measurements take approximately 1 minute including the dead time between each point. Such measurements were made repeatedly by 51 times in 1 hour, and each data was not averaged. As shown below, the revised Figs. S5a, b and d illustrate the absolute values of the electric field trajectory $E_{\text{abs}} = \sqrt{E_x^2 + E_y^2}$ (**a**), the electric field trajectory in the xy plane (**b**), and the power spectra of LCP and RCP components (**d**) for a typical measurement, showing that the measured BCL pulses have the central frequencies of 20 and 30 THz. From these measurements and analysis, we estimated the standard deviation of the azimuthal angle of peak electric field as 14.7 mrad.

It is difficult to evaluate the shot-to-shot stability of azimuthal angles because the POMEOS measurement uses the fitting analysis of modulated signal which is consisted of several pulses. The smallest pulse number for the POMEOS analysis is about 40 in our case, which is decided by the ratio between laser repetition rate and PEM frequency, as discussed in our previous paper ([Ref. 41]). However, we passively stabilize the whole system with temperature-stable air conditioner and an enclosure box for blocking winds. The shot-to-shot pulse energy fluctuation of the compressed NIR pulse is less than 0.04 %.

In the revised manuscript, we added the required information in the Supplementary Information Note 6 and added Fig. S5 (**Change #8**).

Fig. S5. Phase stability of BCL pulses. **a**, Time-dependent fluctuation of the absolute electric field amplitude of BCL pulses (lower) and its single typical measurement (upper). The black arrow shows the direction of the time-dependent fluctuation. **b**, Single typical measurement of the trajectory in the xy -plane. The black arrow shows the direction of the rotational-angle fluctuation. **c**, Azimuthal angles in time-domain (lower) and its typical measurement (upper). **d**, Power spectra of LCP and RCP components of the single typical measurement. **e**, Rotational angle change over 1h measurement.

• Lines 243-244: how is this value estimated? What is the NIR pulse energy associated with such a field? Would it be possible to increase it? What is its main limitation?

For the estimation of the electric field amplitude, we first measured the field of linearly polarized

multi-terahertz pulse. The pulse energy of the linearly polarized multi-terahertz pulse was estimated to be about 0.3 nJ by a power meter (3A-P-THz, Ophir), the beam diameter was 45 μm by a mid-infrared camera (RIGI-S2, Swiss Terahertz), and the pulse width was measured as 19 fs by considering FWHM obtained from Gaussian fitting to the EO sampling signal. From them, the electric field amplitude of the linearly polarized multi-terahertz pulse was calculated to be about 700 kV/cm. Then, assuming the linearity of POMEOS, the electric field amplitude at the peak of the trefoil BCL pulse was estimated to be 100 kV/cm. The signal intensity ratios at the peak of the other BCL pulse are only a few times different from the trefoil BCL pulse.

The main limitation in our current configuration is the NIR pulse energy available in multi-plate compression method. The pulse energy is limited to sub-mJ to avoid the damage in the plates or breakdown process in the air plasma. But the near-infrared pulse energy can be applied to other compression methods such as a hollow core fiber (HCF). For example, the HCF is applied to the similar laser system and sub-2-cycle pulse with 86% efficiency is reported recently [Ref. 44]. We expect that the compressed NIR pulse energy can be increased about several tens of times, and the peak electric field amplitude of the DFG pulse should be reaches to the order of 10 MV/cm.

In the revised manuscript, we added the required information in the Supplementary Information Note 7 (**Change #9**).

• Lines 292-304 and 218-238: the meaning of the jitter of the entire pulse on the t-axis is unclear. How do the authors define a meaningful zero of the time axis? If I understand properly, they retrieve the MIR transient using the POMEOS technique. From Ref. [26] of the manuscript, the time-axis in this two-color measurement corresponds to the delay between the MIR field and a gate pulse. As in any pump-probe experiment, this is not an absolute time axis: shifting arbitrarily both pulses in time, the POMEOS measurement will not change, as it only depends on the gate and MIR delay. Therefore, what is the meaning of this jitter? How can the authors attribute it to a shift of the MIR transient, and not to the gate pulse? How can the timing jitter (and the same applies to the BCL rotation) be distinguished from a change in the carrier-envelope offset of the MIR transient?

The jitter evaluated in this work means the whole jitter in this measurement, which can arise from (i) a jitter between the gate and multi-terahertz pulses and (ii) a fluctuation of the carrier-envelope offset in the multi-terahertz pulse, as expected by Reviewer #1. Distinction of these two origins would be very difficult. As a matter of fact, even if we could suppress one of them, it would not help the whole experiment as long as the other is still large. Therefore, for generating and detecting the BCL pulse, the whole jitter including both origins are the most important value to be suppressed.

In the revised manuscript, we added explanations for the definition of the jitter in the main text and in

the Supplementary Information Note 6 (**Change #10**).

• Figs. 2c1-4, 3a-d, 3e1 and 3f1: the dark blue lines hide some light blue lines. Why are these lines plotted in different colors? Do they have a different meaning? If they are field values where the symmetry of the BCL is not particularly evident, the authors should critically discuss them, explain where they come from and quantify them (e.g., comparing their maximum field value with the peak field amplitude) in the “Discussion” section. In addition, in the *Ex-Ey* plots, using a color scheme where different sections of the line are represented with different colors encoding distinct time values, and using the same color scheme for 3D graphs (Figs. 2a2, 2b2, 3e1, and 3f3) would help understanding how the electric field of the BCL evolves in time.

We appreciate Reviewer #1 for giving this comment. This argument became very important to discuss a potential limitation of our method as follows.

Here we plot the 3D graphs of previous Figs. 2a2, 2b2, 3e1, and 3f3 in the revised Figs. 2 to 3 as shown below. At around the central part of the pulse envelope, the BCL pulse is well controlled as we designed. However, the field trajectory tends to be distorted as it is away from the pulse center. To emphasize the well-controlled part, we used the dark blue lines to be contrasted to the other part shown by the light blue. Compared with the successful generation of the C_3 and C_5 BCL pulses, the C_4 trajectory shows a conspicuous distortion. In other words, the C_4 -like pattern was achieved only in a limited time window. This result originates the set of frequencies n_1 and n_2 at 14 and 39 THz, respectively, which is a bit deviated from the required ratio $n_1:n_2 = 1:3$. We choose these frequencies because the emission at 10-14 THz is weak and not well controlled because of the absorption in the GaSe crystal [J. Opt. Soc. Am. B **26**, A58 (2009)]. Although we have stated that the available bandwidth is 10-40 THz, more strictly speaking, it should be 14-39 THz, and therefore the C_4 -like trajectory is achieved only transiently. This problem is related to a potential limit for the controllability of BCL pulse in our method. To realize the C_n pulse ($n=3, 4, 5, 7, \text{ and } 8$), the required frequency ratio is $n_1:n_2=1:2, 1:3, 2:3, 3:4, \text{ and } 3:5$ respectively. Among them, generating the C_4 BCL pulse requires a pair of largely separated frequency components n_1 and n_2 by a factor of 3, which must be selected in the limited bandwidth. This is the reason why we did not demonstrate the C_6 pulse because it requires $n_1:n_2=1:5$, *i.e.*, the factor of 5 between n_1 and n_2 .

For large numbers of $n>7$, on the other hand, the pair of frequency components are close each other so that many sets of frequencies are available. However, it requires many oscillation cycles to “close” the full trajectory with C_n symmetry. Therefore, generation of the well-designed $C_{n>7}$ BCL pulse requires a certain degree of monochromaticity for each frequency component, which depends on the spectral resolution of the phase control in the $4f$ system with the SLM. By changing the configuration

of $4f$ system, increasing the spectral resolution is possible, but it is accompanied by a decrease of the light intensity.

In summary, the C_4 or C_6 pulse requires a broadband generation of the multiterahertz pulses. The $C_{n>7}$ pulses require a spectral resolution. By contrast, the control of C_3 and C_5 pulses were successfully demonstrated, which will contribute to apply the BCL pulses for controlling the solids.

In the revised manuscript, we explained the definition of the blue and light curves and added the discussion to clarify the potential limitation in the main text (**Change #11**). We also tried to make 3D plots of electric field trajectories and Ex - Ey plots with colors that change with time, as illustrated in Fig. R1 below. However, the electric field trajectories became more complicated, and it looks rather difficult to understand the correspondence and direction of rotation. Thus, we used the original blue color in the revised manuscript.

Fig. 2. Control of shape and orientation of BCL pulse with C_3 symmetry.

Fig. 3. Control of rotational symmetry and helicity of BCL pulses.

Fig. R1. Color scale plot with time change of C_5 -BCL pulse.

- General comment: in my opinion, the manuscript lacks a thorough discussion of the parameters of the generated BCL pulses (energy, duration, pulse contrast, ...), what currently limits them, what is the main limitation to achieving full polarization control over the MIR transient (if there is any), and how to possibly overcome these limitations in the future. I think this information to be crucial for future experimental applications of their approach.

We appreciate the fruitful comment from the Reviewer #1. We added the information of the parameters of BCL pulse including the pulse energy, the duration, and the pulse contrast. For our typical BCL pulse shown in Fig. 2a, we estimated that the pulse energy is 0.2 nJ, the duration is 225 fs, and the pulse contrast is 26 in intensity.

The limitation about the pulse energy is discussed in our reply to the previous comment (page10 and 11). If the much stronger NIR pulse becomes available, the pulse energy might be enhanced by two orders of magnitude. The pulse duration is decided by the designed bandwidth of three bands (U, C, and L). To generate shorter BCL pulse, we should choose broader bandwidth. The limitations of the bandwidth are the overlapping of the bands and spectral structures in the NIR pulse (Fig. 1d). On the other hand, limitation in longer duration is the delay for vanishing the undesired DFG (400 fs in our case). The maximum delay is limited by the resolution of the 4f pulse shaper. The longer pulse is also difficult from the viewpoint of the intensity of DFG. That might be overcome by introducing chirp in three NIR bands, while we used chirp-free pulses in this work. Pulse contrast would be also improved by using the controlled chirps in the spectral components other than three bands to suppress the undesired DFG.

Regarding full polarization control, the chirp control method has recently been demonstrated in THz region [Ref. 45], which might be one of the possible ways. However, in this work we focus on the BCL pulse which has been demanded to realize in the recent theoretical studies. We believe that full polarization control is beyond the scope of this work.

In the revised manuscript, we added the information in the Supplementary Information Note 8 **(Change #12)**.

Minor comments

- Line 34: the authors cite three exemplary experiments, two from the dawn of attosecond science in solids (Refs. [1] and [2] of the manuscript) and one demonstrating Bloch oscillations with HHG in solids (Ref. [3]). Despite their importance, several additional groups have investigated similar phenomena in distinct experimental conditions. To better picture also recent developments on these

topics, and being this an introductory paragraph, I would suggest also citing some recent reviews on these topics (e.g., Refs. [R2]-[R4]).

We appreciate the helpful suggestions. According to the suggestions, we totally reconsidered the list of references and added some important literatures (**Change #13**).

- As a general comment on figures, indicating panels with a letter and sub-panels again with a number (e.g., Figs. 1a1 and 1a2 on line 46) is unusual. My suggestion is to separate the letter from the subsequent number with a dot (e.g., Figs. 1a.1 and 1a.2) for clarity.

We thank Reviewer #1 for a helpful comment. In the revised manuscript, we separate letters from subsequent numbers with dots (**Change #14**).

- Lines 65-66, point (i): in this form, this phrase is not always true. As an example, electron-electron scattering can be extremely fast [R5], even reaching the sub-femtosecond time scale. Electron-phonon scattering, instead, typically takes place on longer time scales. My suggestion is to rephrase it as: “(i) it oscillates faster than typical electron-phonon scattering times [...]”.

We thank Reviewer #1 for a helpful comment. In the revised manuscript, we rephrase as described above (**Change #15**).

- Lines 68-69, point (iii): if I understood the authors’ point properly, they wanted to say that, for the same peak electric field amplitude, multi-THz light fields lead to a larger amplitude of the vector potential compared to visible light. This comes from the inverse scaling of the vector potential amplitude with the frequency. However, increasing the pulse energy or tighter focusing for visible pulses could lead to the same vector potential amplitude. If the authors want to keep this point as an advantage of multi-THz fields, it must be clarified.

Thank you very much for the helpful comment. We found that our explanation was insufficient in the original manuscript. We reply to this comment together with the next one. Please see below.

- Lines 69-70, point (iv): this point is ambiguous. In a two-band system, inter-band excitation can be

categorized by using two adiabaticity parameters [R6]: the ratio between the energy bandgap and the photon energy and the Keldysh parameter. Depending on the experimental conditions, the interaction of light pulses can or cannot lead to a residual excited electron population [R7] and, eventually, damage the material. Thus, at least four quantities must be considered (the electron-hole effective mass, the energy gap, the photon energy, and the peak field amplitude), and two of them depend on the sample considered. Thus, this point must be either clarified, or removed from the discussion.

We totally agree with Reviewer #1 that the mechanism of interband excitation by low-frequency light depends on many parameters. But we will not discuss this point in this work. The reason why we raise the points (iv) in the introduction is that we want to compare the multi-terahertz wave with the near-infrared or visible light where the BCL has already been realized.

For the light wave control of matter, application of strong light field while avoiding the damage and reducing the unwanted electron excitation is essential. For this purpose, using a lower pump photon energy is important to significantly increase the damage threshold, as mentioned in the review article [R2] raised by Reviewer #1. To suppress the unwanted interband electron excitation to avoid the damage, the use of multi-terahertz pulses is much more important than near-infrared or visible pulses, whichever the excitation mechanism by multi-terahertz pulses is the multiphoton process or the Franz-Keldysh-type field-induced ionization.

This is also related to the point (iii) in the last comment. As mentioned by Reviewer #1, a large vector potential is also available in the near-infrared or visible light if the field is strong enough. However, the use of intense near-infrared or visible pulses to the solids tends to easily breakdown a sample due to the interband excitation of electrons, unless ones use wide-gap insulators. By contrast, the recent research interest for applying BCL pulses to condensed matter physics is aimed at the Dirac semimetal [Ref. 30], graphene [Ref. 29], or other materials with a small band gap or even a gapless band structure. Therefore, ones have to avoid the intense high-frequency field.

To improve the statement in the introduction, we combined the points (iii) and (iv), and clearly mentioned the comparison between multi-terahertz pulses and near-infrared or visible pulses in the revised manuscript (**Change #16**).

• Lines 96-104: the authors discuss the energy and angular momentum conservation that is crucial for their scheme. For light pulses, they only relate the angular momentum to the polarization state of light. I would thus suggest replacing “angular momentum” with “(spin) angular momentum” when it is referred to the properties of the light beam.

According to the suggestion, we modified it in the revised manuscript (**Change #17**).

- Lines 125-126: why is the phase modulated along the azimuthal angles of $\pm 45^\circ$? Is there some physical reason for this? Is it connected with the implementation of the SLM?

The reason for the phase modulation along $\pm 45^\circ$ is that this is the most typical configuration of the commercially available dual SLM, where the phase of light can be controlled along with the two directions by using two nematic liquid crystals. It is natural because, before the input of SLM, the light is linearly polarized horizontally (0°) or vertically (90°), which behaves as *p*- or *s*-polarized light when it is reflected by mirrors on an optical table. The well-defined *p*- or *s*-polarization is quite important to design the optical system. To control the horizontally (0°) or vertically (90°) polarized light, the phase modulations along $\pm 45^\circ$ directions are the most natural directions.

We added a brief explanation with a reference to SLM in the revised manuscript (**Change #18**).

- Line 179: while in the discussion of Fig. 1 the authors refer to the amplitude of the ω and 2ω light fields, here and in Fig. 2 they present an intensity ratio. Even though the two quantities are clearly connected, probably choosing only one for the whole manuscript would make it clearer. In addition, an intensity ratio of 0.2 gives an amplitude ratio of $\sqrt{0.2} \approx 0.45$. Repeating the simulation of Fig. 1a2 with the same ratio as the experiment would allow a more direct comparison.

We choose the intensity ratio in the whole manuscript (**Change #19**).

- Line 187: the authors discuss the orientation control of the BCL trajectory. If this type of control is general and valid for all the trajectories in Fig. 1a1, 1a2, and 1a3, the authors should clearly state it and demonstrate it with supplementary data. If instead it is specific of the trajectory in Fig. 1a1, they should clearly state it.

We confirmed that the orientational control is possible for other trajectories as shown in Fig. S2 in the revised manuscript. In this revision, we showed these data to demonstrate the orientational control of other trajectories in the Supplementary Information Note 3 (**Change #20**).

Fig. S2. Control of the orientation of BCL pulses with C_3 and C_5 symmetries. A–d, the orientation control of triangle-like BCL trajectory. **e–h**, the orientation control of star-like BCL trajectory.

- Line 196: applying the offset phase α to the L-band would have given the same result? Would the formula on line 198 remain the same, but switching R with L and vice versa? If this is the case, then in the example of Fig. 1a1 $\omega_L = 2\omega$ and $\omega_R = \omega$, so for the same α the rotation should be larger.

Applying α to the L -band gives the BCL rotation of $-\omega_L \alpha / (\omega_L + \omega_R)$ by switching R with L. In the revised manuscript, we added a brief explanation in the Supplementary Information Note 2 (**Change #21**).

- Line 221-222: as correctly highlighted, in the 4f setup different components follow different optical paths. Have the authors quantified the possible timing jitter between the different (U, C, L) bands? Does the setup require any active or passive stabilization scheme?

As discussed in Fig. S5e above, the phase fluctuations of ω_R and ω_L components reflect the timing jitter between different bands. We have estimated that the timing was the order of 0.1 fs in one-hour measurement. The timing jitter might be caused by air fluctuation, thermal expansion, and other reasons. To suppress them, we employed passive stabilization in 4f setup, covering by a box. The temperature stability was less than 0.1°C, while the possible timing jitter from thermal expansion is

estimated as 46 fs/°C. Comparing with our experimental results (the order of 0.1 fs), the thermal expansion effect is well suppressed in our setup. Here, we can estimate the timing jitter as $\Delta t = \xi_{Al} L_0 \Delta\theta / c$, where $\Delta\theta$ is the temperature change, $\xi_{Al} = 23 \mu\text{m}/(\text{m}\cdot\text{K})$ is the coefficient of aluminum thermal expansion, $L_0 = 0.6 \text{ m}$ is the path length of the 4f optical system, and c is the speed of light. Note that we do not use active feedback in the 4f setup itself, but before that, we used only one active feedback system to stabilize the input beam position to the multiplate broadening. As long as the beam for the 4f setup is stable enough, we do not need active feedback in the 4f setup itself for generating stable BCL pulses as demonstrated here.

Though there exists the timing jitter between different bands, we confirmed that most of them are canceled in BCL pulses. We measured the jitter of trefoil BCL pulses as shown in Fig. S5e, which are evaluated sufficiently stable.

In the revised manuscript, we added the explanation of our system in Supplementary Information Note 6 (**Change #22**), and brief explanation about the feedback of multi-plate broadening in the Methods section (**Change #23**).

• Line 238: a beam path difference of 60 nm corresponds to a timing jitter of ~200 as. In attosecond science, this mechanical stability is hard to achieve without any mechanical feedback (see for example blue line in Fig. 5a from [R8], where the phase error in the open loop condition corresponds to a phase delay shift of several femtoseconds over less than one hour). Can the authors comment one on how they achieved such an impressive stability? What is the length of the 4f setup?

The length of the 4f setup is 60 cm. To achieve the stable measurement, we carefully covered the whole 4f setup by using a black polypropylene to protect it from the temperature fluctuation in environmental atmosphere.

We added the information in the Supplementary Information Note 6 (**Change #22**)

• Line 258: a more quantitative value on the pulse duration is required. What is its exact value? Is it expressed as the intensity full width at half-maximum duration? How is it characterized? What is the associated Fourier transform limited duration?

As pointed out here, we noticed that the expression “less than 12 fs” might be ambiguous. We change the description to “approximately 12 fs” in the revised manuscript.

As Reviewer #1 expected, the pulse duration is defined as the full width at half-maximum for the intensity profile, which was evaluated by second harmonic (SHG) frequency-resolved optical gating

(FROG) method. The typical results are shown in Fig. S4, where the pulse duration is 11.9 fs as illustrated with black arrows in Fig. S4c. However, the pulse duration slightly fluctuates day by day, between 11 and 12 fs. Therefore, we use the expression “approximately 12 fs” in the revision.

The Fourier limit pulse also is shown in Fig. S4c, whose duration is 11.7 fs. The measured pulse duration is well close to that of the Fourier limit pulse.

In the revised manuscript, we added the detail of the evaluation of the compressed pulse in Supplementary Information Note 5 (**Change #24**).

Fig. S4. FROG measurement of NIR compressed pulse. **a**, Measured and retrieved SHG-FROG traces after the output of the multi-plate pulse broadening. **b**, Retrieved intensity spectrum (red) and spectral phase (blue). **c**, Time-domain retrieved intensity envelope (red) and its Fourier transform limit pulse (blue). The black arrows show the width of the FWHM.

- Lines 261-263: including a reference to published literature describing the basic working principle of the SLM in the $4f$ configuration would expand the readership of the manuscript.

In the revised manuscript, we added references [Refs. 45 and 46] to the basic principle of the SLM in the $4f$ configuration (**Change #18**).

- Line 268: is the concave mirror a spherical mirror? What is the focal length?

The concave mirror is a spherical mirror with a focal length of 150 mm.

In the revised manuscript, we added this information in the Methods section (**Change #25**).

- Line 287: are the MIR and NIR beams impinging orthogonally on the Ge filter? What is its thickness?

The thickness of the Ge filter is 0.5 mm, which is thick enough to totally cut the near-infrared pulse.

By contrast, the filter is transparent for the multi-terahertz pulse at 10-40 THz (40-160 meV).

In the revised manuscript, we added a brief explanation in the Methods section (**Change #26**).

- Despite making use of a stretchable hollow-core fiber, the main point of Ref. [27] is the generation of soft x-rays in the water window via high-order harmonic generation (HHG). Ref. [R9], being a review on the generation of high-energy few-cycle laser pulses, is probably better.

According to the suggestion, we modified the reference (**Change #27**).

- Figs. 2c1-4: how is the red dashed line obtained? If it is just a guide to the eyes, it should be stated in the caption.

It is not a guide to the eyes. The line shows the angle which we set in the SLM on software.

In the revised manuscript, we clearly stated the definition of the red line in the main text and the figure caption (**Change #28**).

- Figs. 3a-d: strictly speaking, the frequency ratio in the bottom right corner (e.g., 17:34 in Fig. 3a) is a dimensionless quantity. I suggest to not express this quantity as a ratio.

As pointed out by Reviewer #1, the frequency ratio is not a good expression.

In the revised manuscript, we modified it to another style, e.g., 17 THz + 34 THz in Fig. 3a and also added an explanation in the figure caption (**Change #29**).

Typos and suggestions

- Lines 42-43: trajectories of the light field vector → trajectories of the light electric field vector.
- Lines 44-45, “Counterrotating [...] delineate”: this phrase is difficult to read. I suggest rewriting it as: “Counterrotating bicircular light fields (BCL) with frequencies ω (fundamental) and 2ω (second harmonic) delineate [...]”.
- Line 136: the centre frequency of the C-band → the central frequency of the C-band.
- Line 201: Third, order n → Third, the order n .
- Line 208-209: the rotating direction of the trajectory → the direction of rotation of the trajectory.
- Line 256: Of the output, 11% (0.21 mJ) → 11% of the output (0.21 mJ).
- Dashes are missing in the titles of refs. [3] and [13].

We thank to all the suggestions by Reviewer #1. We revised all the typos above in the revised manuscript. **(Change #30)**

- Figs. 2a1, 2b1, 3e2 and 3f2: using a different line type (e.g., dashed) for LCP or RCP would make the difference more evident even in printed black and white versions of the manuscript.

We agree that LCP or RCP components in Figs. 2a1, 2b1, 3e2 and 3f would make a difference. However, we consider that changing the shape of the lines (e.g., dotted lines) would be undesirable because it would make the LCP and RCP elements seems to be asymmetrical. Therefore, we changed the pattern of the mesh between LCP and RCP due to be distinguished even in monochrome. **(Change #31)**

- Figs. 2a2, 2b2, 2c1-4, 3a-d, 3e1, 3e3, 3f1 and 3f3: I suggest normalizing the field amplitude in all panels (e.g., to the maximum field amplitude in the transient). Moreover, panels showing the same quantity, as 3e1 and the right-hand side of 3e3, should have the same normalization.

According to the suggestions, we normalized all the figures pointed by Reviewer #1 to the maximum field amplitude (**Change #32**).

Reply to Report of Reviewer #2

Report and our reply are shown in cyan and black, respectively. The reference numbers below are those updated in the revised manuscript.

The paper is entitled "Programmable generation of counterrotating bicircular light pulses in the multi-terahertz frequency range" and it explores the generation of counterrotating bicircular light pulses in the mid-infrared or multi-terahertz region. The paper introduces a methodology for generating phase stable counterrotating bicircular light within the 10-40 frequency band. This methodology is achieved with an slm and intra-pulse difference frequency generation.

The results presented in the paper are certainly intriguing and hold potential significance for the community interested in ultrafast science and condensed matter physics. However while the results are certainly intriguing I have reservations regarding the novelty of the paper. In particular polarization control of bicircular light has been demonstrated previously. In particular in the article (<https://doi.org/10.1364/JOSAB.456066>). However the following articles is also very relevant to the paper. (<https://doi.org/10.3390/photonics10070803>). It is noteworthy that both of these articles have been omitted by the authors. Without the novelty claim on the polarisation control of bicircular light, I do not believe that being programmable and not requiring waveplates or polarizers is a sufficient claim for publication within nature communications.

We acknowledge Reviewer #2 for reviewing our manuscript and highly evaluating the potential significance of our work.

Here we explain two literatures raised by Reviewer #2.

(1) <https://doi.org/10.1364/JOSAB.456066>

Although Reviewer #2 introduced this work as an article to demonstrate the polarization control of bicircular light, it is not correct. This work reports a theoretical calculation for a polarization control of a THz pulse by using **co-rotating** two-color circularly polarized light in the near infrared. Note that they did not discuss the polarization control of bicircular light itself. Moreover, the trajectory of electric field vector in such a **co-rotating** bicircular light does not draw a trefoil-like pattern and totally different from the **counter-rotating** bicircular light discussed in this work.

(2) <https://doi.org/10.3390/photonics1007080>

This article also reports the result of theoretical calculation aimed for a polarization control of a THz pulse, and even do not use the bicircular light.

Because these two literatures are not related with our work at all, we concluded that they are not to be cited in our paper. **However, we agree with Reviewer #2 that we should improve our manuscript to properly describe the novelty of this work with comparing previous works.**

In this major revision of the manuscript, we largely modified the introduction part (**Change #33**), and also added proper references (**Change #13**) to clarify the novelty of this work compared with previous literature.

1). Describing something in S^{-1} is highly unusual, especially when you frequently refer to it as terahertz, why not just use Hz?

We used “s⁻¹” only twice in the introduction part as “terahertz (10^{12} s⁻¹) or petahertz (10^{15} s⁻¹) electric fields.” This is just a paraphrase expression to refer to the light field for the broad readership of *Nature Communications*. We believe that it is not unusual.

2). In my opinion "intra-pulse difference frequency generation (DFG)" should be modified to "optical rectification" (a well studied phenomena under this name) to better align to the nomenclature used by the terahertz community.

We agree with Reviewer #2 that “optical rectification” is frequently used in the terahertz community. According to the basic textbooks of nonlinear optics [*e.g.*, R. W. Boyd, “Nonlinear optics” and Y. R. Shen, “The Principles of Nonlinear Optics”], the definition of “optical rectification” is to generate a zero-frequency (static) electric field by second-order nonlinear interaction using exactly the same frequencies: $E(\omega)$ and $E(\omega)^*$. The conventional THz wave around 1 THz could often be regarded as the DC-limit electric field, which justifies ignoring the finite frequency of the THz pulse. In the case of our work, however, we cannot regard the generated multi-terahertz pulse as a static field because it must oscillate in time to produce BCL. In this sense, the generation process is better described by using the term “difference frequency generation” because it is a more generalized second-order nonlinear interaction using slightly different frequencies: $E(\omega_1)$ and $E(\omega_2)^*$. For the case of using two frequency components ω_1 and ω_2 in a single laser pulse, the term “intra-pulse difference frequency generation” has often been used to describe the multi-terahertz or midinfrared pulse; please see *e.g.*, [<https://doi.org/10.1038/s41377-018-0099-5>] and many other literatures.

REVIEWERS' COMMENTS

Reviewer #1 (Remarks to the Author):

Reviewer report for manuscript ID: NCOMMS-24-04948A

Title: Programmable generation of counterrotating bicircular light pulses in the multi-terahertz frequency range

Authors: Kotaro Ogawa, Natsuki Kanda, Yuta Murotani, and Ryusuke Matsunaga

The manuscript is a revised version of NCOMMS-24-04984. The authors have extensively and adequately addressed all my concerns, expanded significantly both the manuscript and the methods section, and included a supplementary information file. In my opinion, these changes led to an overall improvement of the manuscript, which I now consider ready for publication in Nature Communications. As a final remark, I would like to point out the following typos and suggestions:

- Line 77: [...] fields³⁶; Otherwise [...] → [...] fields³⁶; otherwise, [...].
- Line 175: [...] of its parameter → [...] of its parameters.
- Line 328: [...] condition, the higher-frequency range of generated [...] → [...] condition. The higher-frequency range of the generated [...].
- Line 329: Furthermore, higher frequency side [...] → Furthermore, the higher frequency side [...].
- Lines 342-343: if I understood properly, “system” and “environment” respectively indicate the photons (regardless of their frequency) and the crystal. I would suggest clarifying these expressions.
- Line 389, ref. 1: the correct title is “Optical-field-induced current in dielectrics”.
- Lines 395-396, ref. 3: is the link to the supplementary information of Schubert et al. included on purpose?
- Lines 420 and 427-428, refs. 15, 18-19: title missing.
- Line 524: (a1) a trefoil [...] → (a1) A trefoil [...].
- Line 539, “b, Corresponding results for different amplitude ratio [...]”: it might be helpful to state this amplitude ratio explicitly also in the caption.
- Line 543, “[...] and 90° which we set [...]”: space missing before “which”.
- Line 142 of Supplementary Information: [...] are centered frequencies [...] → [...] are centered at frequencies [...].

Reviewer #2 (Remarks to the Author):

The authors have made substantial changes to their work, and while my initial review focused on the author's similarity with previous works. They have clarified the difference and performed all necessary changes for this manuscript to be published, any of the differences that were not changed are creative differences and I am happy to recommend publication.

Reply to Report of Reviewer #1

Reviewer report and our reply are shown in cyan and black, respectively.

The manuscript is a revised version of NCOMMS-24-04984. The authors have extensively and adequately addressed all my concerns, expanded significantly both the manuscript and the methods section, and included a supplementary information file. In my opinion, these changes led to an overall improvement of the manuscript, which I now consider ready for publication in Nature Communications.

We highly appreciate Reviewer #1's excellent efforts, which greatly assisted us in enhancing the manuscript. We are glad to know that Reviewer #1 esteemed our revised manuscript. We also thank for giving some additional suggestions. We made minor revisions as follows.

- Lines 342-343: if I understood properly, “system” and “environment” respectively indicate the photons (regardless of their frequency) and the crystal. I would suggest clarifying these expressions.

As reviewer #1 pointed out, the definition of “system” and “environment” is ambiguous. We rephrased the sentence as below. **(Change #1)**

Therefore, an excess angular momentum of $\pm\hbar$ or $\pm 3\hbar$ should be transferred to the crystal, which means that DFG is allowed only for $n=1$ or 3.

- Line 539, “b, Corresponding results for different amplitude ratio [...]”: it might be helpful to state this amplitude ratio explicitly also in the caption.

We added the intensity ratio in the caption, where we rephrase from “amplitude ratio” to “intensity ratio” due to unify the expression to the main manuscript. **(Change #2)**

Typos and suggestions

- Line 77: [...] fields³⁶; Otherwise [...] → [...] fields³⁶; otherwise, [...].
- Line 175: [...] of its parameter → [...] of its parameters.
- Line 328: [...] condition, the higher-frequency range of generated [...] → [...] condition. The higher-

frequency range of the generated [...].

- Line 329: Furthermore, higher frequency side [...] → Furthermore, the higher frequency side [...].
- Line 389, ref. 1: the correct title is “Optical-field-induced current in dielectrics”.
- Lines 395-396, ref. 3: is the link to the supplementary information of Schubert et al. included on purpose?
- Lines 420 and 427-428, refs. 15, 18-19: title missing.
- Line 524: (a1) a trefoil [...] → (a1) A trefoil [...].
- Line 543, “[...] and 90° which we set [...]”]: space missing before “which”.
- Line 142 of Supplementary Information: [...] are centered frequencies [...] → [...] are centered at frequencies [...].

We are thankful to all the suggestions by Reviewer #1. We revised all the typos above in the revised manuscript. **(Change #3)**

Reply to Report of Reviewer #2

Report and our reply are shown in cyan and black, respectively.

The authors have made substantial changes to their work, and while my initial review focused on the author's similarity with previous works. They have clarified the difference and performed all necessary changes for this manuscript to be published, any of the differences that were not changed are creative differences and I am happy to recommend publication.

We thank Reviewer #2 for spending time on the reviewing our manuscript. We appreciate for evaluating our revision and thank to the recommendation of publication.

List of changes

1. We revised a sentence about excess angular momentum transfer in the Methods section.

2. We added the intensity ratio explicitly in the caption of Fig. 2.
 3. We corrected the typos.
-